# CXCL12 targets the primary cilium cAMP/cGMP ratio to regulate cell polarity during migration

Melody Atkins [1] ✉, Maud Wurmser[2], Michèle Darmon[1], Fiona Roche[2], Xavier Nicol [2] & Christine Métin [1] ✉

Directed cell migration requires sustained cell polarisation. In migrating cortical interneurons, nuclear movements are directed towards the centrosome that organises the primary cilium signalling hub. Primary cilium-elicited signalling, and how it affects migration, remain however ill characterised. Here, we show that altering cAMP/cGMP levels in the primary cilium by buffering cAMP, cGMP or by locally increasing cAMP, influences the polarity and directionality of migrating interneurons, whereas buffering cAMP or cGMP in the apposed centrosome compartment alters their motility. Remarkably, we identify CXCL12 as a trigger that targets the ciliary cAMP/cGMP ratio to promote sustained polarity and directed migration. We thereby uncover cAMP/cGMP levels in the primary cilium as a major target of extrinsic cues and as the steering wheel of neuronal migration.

Neuronal migration represents a major phase of brain development and abnormal migration has been linked to several neurological and mental disorders. Cell migration is defined as a directed motility process, and neurons, which are highly polarised cells, often progress in a saltatory way sometimes called migration by nucleokinesis[1]. They first extend a long leading process in a permissive or attractive environment. The centrosome moves forward to a proximal region of this process, known as the dilatation or swelling compartment[2,3]. Thereafter, the nucleus dynamically translocates towards the centrosome and the nuclear and swelling compartments merge together, allowing a new cycle to start over. Interestingly, the centrosome is also known as the organiser of the primary cilium (PC), a small microtubule-based structure that extends at the surface of almost all vertebrate cells. Ciliogenesis first involves the docking of a modified mother centriole−the basal body−to the plasma membrane. Extension of the ciliary membrane and microtubule core−or axoneme−is then ensured by a process termed intraflagellar transport, which involves the molecular motor-based bidirectional transport of various ciliary components[4]. Despite the structural continuum between the centrosome basal body and the ciliary axoneme, a transition zone assembled at the base of the PC acts as a physical barrier separating the cytoplasm from the clioplasm. Remarkably, the PC has been involved in the long distance migration of neural crest cells[5] and tangentially migrating interneurons[6,7]. Like the centrosome, the PC moves forward to the swelling compartment[6,8]. Migrating interneurons have been shown to assemble a highly dynamic primary cilium[6,7], that hosts membrane receptors for guidance cues known to instruct migration[7]. Although guidance receptors are generally assumed to collect information at the cell front to guide growth cones in the environment, their potential role in the regulation of neuronal migration when located at the opposite pole of the cell within the PC remains unexplored. Our lab showed that PC ablation in migrating interneurons prevents their reorientation towards their final target[6], but the mechanisms responsible for this abnormal migratory behaviour have not been identified.

Long considered as a vestigial organelle of little functional importance, the PC is the target organelle of a family of developmental disorders termed ciliopathies[9], and is now well established as an antenna-like signalling hub, concentrating many specialised signalling components[4,10], among which components of the Hedgehog (Hh), WNT, receptor tyrosine kinase, transforming growth factor β (TGF-β), or bone morphogenetic protein (BMP) pathways. Primary cilia are also

[1]INSERM UMR-S 1270; Institut du Fer à Moulin, Sorbonne Université, F-75005 Paris, France. [2]Institut de la Vision, Sorbonne Université, INSERM CNRS, F-75012 Paris, France. ✉e-mail: melody.atkins@inserm.fr; christine.metin@inserm.fr

specialised compartments for second messenger signalling, such as signalling through the cAMP[11–14] and cGMP cyclic nucleotides[15]. Neuronal primary cilia are no exception. They indeed harbour the AC3 adenylyl cyclase[16], which produces cAMP, various GPCRs, responsible for the inhibition or activation of adenylyl cyclase-mediated cAMP production[17], and some of the main cAMP downstream effectors, such as the PKA kinase[18]. Similarly, cGMP signalling components have been reported in the outer segment photoreceptor[19] or in ciliary compartments of *C. Elegans* neurons[20,21]. However, the physiological relevance of ciliary cAMP and cGMP signals in neurons is still lacking to complete the jigsaw, especially in a context of migration. Remarkably, some of the above-mentioned guidance cues–i.e., semaphorin and CXCL12– required for accurate cortical interneuron migration have been reported to converge onto these second messengers in a PC-independent context[22–26]. This therefore opens the possibility of a role for PC-elicited cAMP and cGMP signals downstream of extracellular guidance cues in order to instruct cortical interneuron migration.

cAMP and cGMP function together during different biological processes, whether in a converging[27,28] or antagonising mode[29–32]. cGMP and cAMP for example exert an antagonistic effect on axono- versus dendritogenesis in hippocampal neurites[32], or on attraction versus repulsion in response to Netrin-1 in neuronal growth cones[31]. Of note, they have both been involved in cortical interneuron migration[22,23,33], but only independently of each other. Moreover, their role in migration has exclusively been assessed at the whole-cell level. This whole-cell approach is in contradiction with an increasing amount of data pointing at a subcellular segregation of cAMP and cGMP micro- domains in order to spatiotemporally orchestrate various cellular responses[12,29,34,35]. The many mechanisms required for accurate migration may thus be spatially organised and integrated within separate and specific subcellular compartments.

Here, we addressed this conceptual challenge by tackling a technical one: the targeting of signalling molecules within subcellular structures. We have developed an innovative toolset allowing the specific and local modulation of cAMP or cGMP levels within the PC of migrating cortical interneurons. We first show that specific ciliary cAMP or cGMP buffering dysregulates the cell polarity and directed migration of in vitro migrating interneurons in an opposite manner. We further demonstrate that these phenotypes are specific to the PC, since targeting our scavengers to a neighbouring subcellular com- partment, i.e., the centrosome, no longer affects cell polarity, but rather cell motility. Photo-activation experiments moreover reveal that increasing ciliary cAMP levels phenocopies ciliary cGMP buf- fering, suggesting a mechanism by which opposite ciliary second messenger levels induce opposite cell polarity regulation. Finally, by combining in vitro pharmacological and ex vivo approaches in grafted brain organotypic slices, we propose a model in which the CXCL12 chemokine secreted by cortical progenitors – which pro- motes the highly directional tangential migration of interneurons in vivo – targets the ciliary cAMP/cGMP levels of cortical inter- neurons and functions as an ON/OFF switch to set their highly directional mode of migration.

## Results

### Buffering cGMP or cAMP in the PC affects the polarity of migrating cortical interneurons

In order to assess the role of cAMP or cGMP signals in the PC of migrating cortical interneurons, we buffered cGMP or cAMP signals locally and specifically within the PC of migrating cells. This was achieved by addressing genetically encoded chelators specific of cGMP[36] (SponGee) or cAMP[37,38] (cAMP Sponge) to the PC by fusion to the 5HT6 targeting sequence[11], which codes for a ciliary G protein- coupled receptor[7]. Scavengers were moreover fused to the mRFP reporter to assess their subcellular localisation (Fig. 1A). The mRFP- tagged 5HT6 sequence–devoid of any of the two sponges–was used as

a control construct. All constructs were co-electroporated in MGE- derived cortical interneurons with a cytoplasmic GFP reporter to monitor cell morphology. Efficient PC targeting of the 5HT6-SponGee- and 5HT6-cAMP Sponge-encoded proteins was confirmed by co- localisation with the Arl13b marker in immunohistochemistry experi- ments (Fig. 1B, C).

Using an in vitro co-culture model previously established in the lab to analyse the motility and directionality of electroporated MGE- derived interneurons, we analysed the consequences of ciliary cGMP or cAMP buffering on such migratory behaviours. Electroporated E14.5 medial ganglionic eminence (MGE) explants were co-cultured and allowed to migrate on a substrate of dissociated cortical cells (Fig. 1D). Time-lapse confocal imaging was then performed as MGE- derived interneurons started to migrate out of their explant and onto the cortical substrate (Fig. 1E). Of note, electroporated migrating interneurons display a highly dynamic mRFP-positive primary cilium, which they rhythmically extend and retract (Supplementary Movies 1 & 2). Only ciliated cells – i.e., expressing both the cytoplasmic GFP construct and mRFP-tagged ciliary construct – were taken into account for the analyses. In this well-characterised in vitro preparation[2,6] (Fig. 1D), control interneurons migrate along a pretty straight trajectory, in which changes of polarity and direction – although possible – remain marginal (Fig. 1D-E). We first sought to rule out a potential involvement of the previously described 5HT6 constitutive activity – leading to increased cAMP production via Gαs signalling[39,40] – in the directionality and motility parameters of electroporated MGE cells. The migration of 5HT6-electroporated MGE cells was thus compared to cells electroporated with a mutated form of the 5HT6 receptor, devoid of any constitutive activity[40] (Gs- dead mutation; Fig. 1E, F). Remarkably, the mean directionality ratios of migrating interneurons, as well as their mean migration speed, are unaffected between the two conditions (Fig. 1I-K), validating our approach and confirming that neither cell directionality nor motility are altered by the 5HT6 receptor.

We thus proceeded with the analysis of migratory behaviours when addressing the cGMP and cAMP scavengers to the PC of migrating cells via the 5HT6 receptor. Of note, neither of the two ciliary sponges prevents migration (Fig. 1G-H; Supplementary Movies 1 & 2). Compared to control cells, ciliary cGMP buffering reduces the mean directionality ratios at each time point (Fig. 1G, L-M), in association with a reduced duration of persistent migration compared to control cells (Fig. 1N), due to frequent changes in polarity (Supplementary Movie 1). As a direct consequence of such changes in polarity – and thereby in directionality –, the average migration speed is decreased compared to 5HT6-electroporated cells (Fig. 1O). By contrast, target- ing cAMP Sponge to the PC leads to highly directional cells with increased directionality ratios at each time point compared to controls (Fig. 1H, P-Q; Supplementary Movie 2), and no change in the average migration speed (Fig. 1R).

Changes in cortical interneuron directionality during their journey to the developing cortex have been linked to changes in branching behaviours[22,23,41]. Two major branching mechanisms exist. While exploring their environment for guidance information, inter- neuron growth cones do not turn, but rather split, forming a dichotomised leading process. Only one of the two end branches is stabilised, while the other retracts before nucleokinesis, usually resulting in mild adjustments of the migration direction. Alter- natively, new branches can be formed from the soma/swelling compartment, allowing the centrosome to select a new leading process in which the nucleus migrates, while the former leading process retracts. Since the nucleus-centrosome axis defines the cell polarity, this second branching mechanism induces drastic changes in polarity and migration direction, comprising polarity reversals. We thus analysed the branching phenotype of migrating cells by distin- guishing branching events observed on the leading process or at the

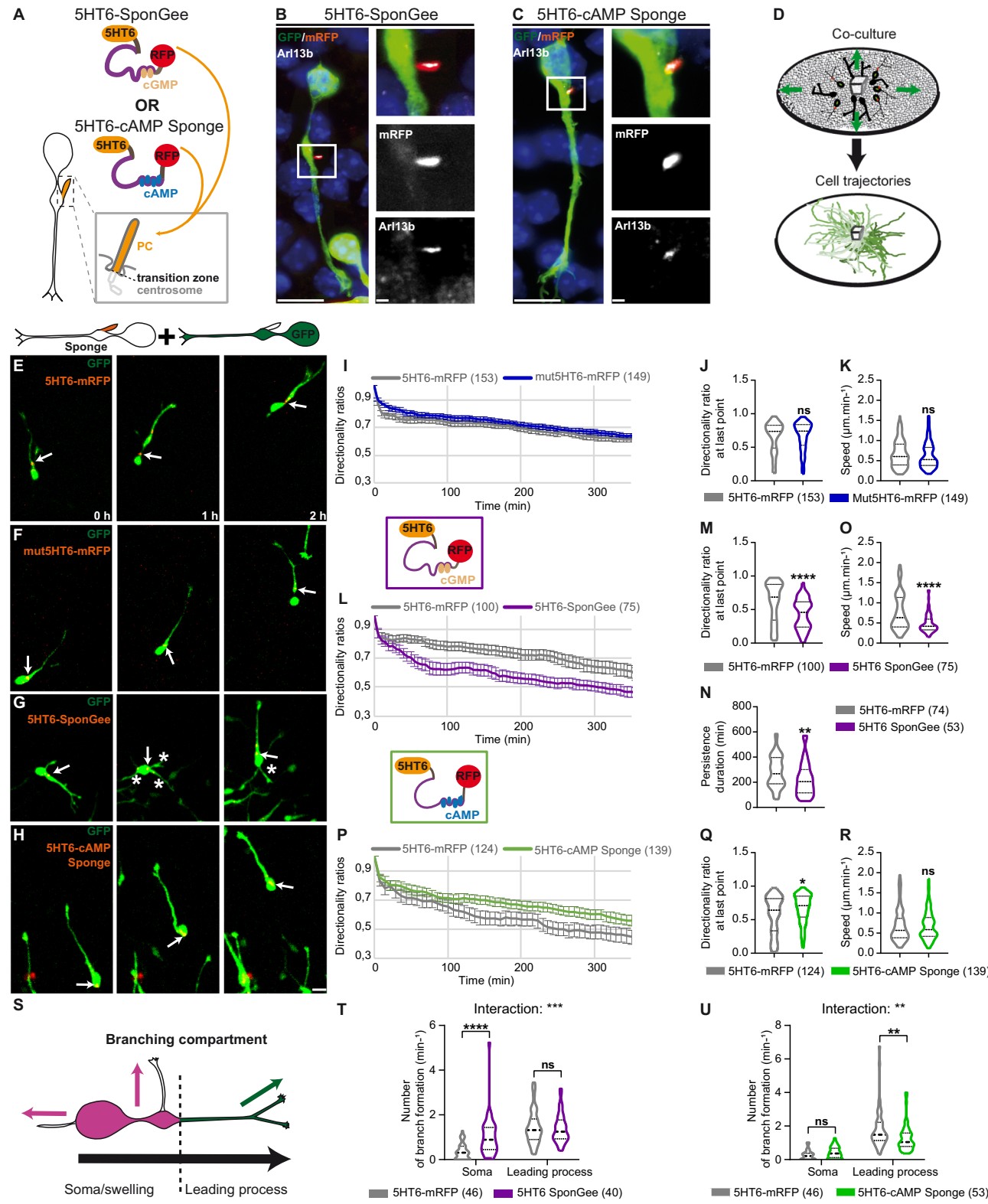

soma/swelling compartment (Fig. 1S). Our results show an interaction in two-way anova tests between the genotype and branching compartment following the electroporation of either of the two ciliary-targeted scavengers (Fig. 1T-U). Ciliary cGMP buffering increases branching at the soma/swelling compartment compared to controls, while branching at the leading process is unchanged (Fig. 1T), which is coherent with reduced directionality ratios (Fig. 1L). Conversely, for ciliary cAMP buffering, branching at the

soma is unchanged, while branching frequencies along the leading process decrease compared to controls (Fig. 1U), in agreement with more directional trajectories (Fig. 1P).

Taken together, our results show that buffering cGMP or cAMP at the PC of in vitro migrating cortical interneurons leads to an opposite dysregulation of cell polarity. While ciliary cGMP buffering induces frequent changes in polarity, ciliary cAMP buffering is responsible for a polarity maintenance phenotype.

**Fig. 1 | Specific buffering of ciliary cGMP or cAMP signals impairs cortical interneuron migratory behaviours in an opposite manner. A** Cortical interneuron PC, organised by the centrosome and separated from the cytoplasm by the transition zone. The mRFP-tagged SponGee or cAMP Sponge are fused to 5HT6 for PC targeting. **B, C** High magnification of cortical interneurons co-electroporated with a cytoplasmic GFP construct and 5HT6-SponGee (**B**) or 5HT6-cAMP Sponge (**C**). Immunostaining with anti-GFP, anti-RFP and anti-Arl13b antibodies revealed the efficient co-localisation of the mRFP-tagged sponges with Arl13b. Insets are higher magnifications of the boxed region on the left. Scale bar, 5 μm; in insets, 1 μm. Co-localisation was observed in three independent experiments. **D** MGE explant co-cultured on a dissociated cortical substrate. MGE-derived cortical interneurons display individual trajectories that radiate away from the MGE explant. **E–H** Time-lapse recordings of cortical interneurons co-electroporated with the GFP cyto-plasmic construct and the control mRFP-tagged 5HT6 (**E**) and mut5HT6 constructs (**F**), 5HT6-SponGee (**G**, see Supplementary Movie 1) or 5HT6-cAMP Sponge (**H**, see Supplementary Movie 2). Arrows and asterisks point at the dynamic mRFP-tagged

PC and branch formation at the soma, respectively. Scale bar, 10 μm. **I–R** Mean directionality ratios at each time point (**I, L, P**) or after a maximum 350-minute migration period (**J, M, Q**), mean persistence duration (**N**) and mean migration speed (**K, O, R**) measured between the 5HT6 and the mut5HT6 (**I–K**), 5HT6-SponGee (**L–O**) or 5HT6-cAMP Sponge conditions (**P–R**). **S** Schematics of branch quantification in (**T, U**). Leading process branches (green) initiate milder changes in direction (green arrow) than branches from the soma/swelling (pink; pink arrows). **T, U** Mean branching frequency from the soma/swelling and leading process for 5HT6-SponGee- (**T**) and 5HT6-cAMP Sponge-electroporated cells (**U**) compared to 5HT6 controls. The number of cells is indicated in the graph legends. *, P ≤ 0.05; **, $P \le 0.01$; ***, $P \le 0.001$; ****, $P \le 0.0001$, ns, non significant. Two-tailed Mann–Whitney test (J, K, M, N, O, Q, R). Two-way ANOVA test with Bonferroni's multiple comparison post test (**T–U**). (**T**) Interaction: ***; Genotype effect: **; Compartment effect: ****. (**U**) Interaction: **; Genotype effect: ns; Compartment effect: ****. Error bars are SEM. Source data and *p* values are provided in the Source data file.

## Ciliary cGMP and cAMP-dependent migratory defects are specific to the PC compartment

We then asked the question of whether cAMP and cGMP signals regulate the same cellular function(s) at the centrosome and at the PC of migrating interneurons. Indeed, the PC and centrosome contribute to a functional unit made of two physically linked compartments that are nevertheless separated by a barrier, the transition zone (Fig. 2A).

To answer this question, we addressed SponGee and cAMP Sponge to the centrosome by fusion to the PACT targeting sequence[42] (Fig. 2A–C). Remarkably, and in contrast to PC targeting, both scavengers have the same effect on cortical interneuron migration when addressed to the centrosome. Compared to cells electroporated with the control PACT-mRFP construct – devoid of any sponge –, neither of the two scavengers (Fig. 2D–F) has an effect on cell directionality ratios (Fig. 2G–H, M–N) or on the duration of persistent migration (Fig. 2I, O). This suggests that unlike at the PC, cGMP or cAMP buffering at the centrosome does not have an effect on cell polarity. Unlike at the PC, the decreased migration speed we observe compared to controls (Fig. 2J, P) is therefore not a consequence of changes in polarity or direction. Further analyses show that in this case, decreased migration speed is linked to an increase in the mean duration of pauses (Fig. 2K, Q) and a decrease in the frequency of nucleokinesis (Fig. 2L, R). Our results therefore support a model in which centrosome-located cGMP and cAMP signals regulate nucleokinesis during cortical interneuron migration, while regulating cell polarity when located within the PC subcellular compartment (Supplementary Fig. 1).

To confirm the specificity of the phenotypes obtained when SponGee and cAMP Sponge are targeted to the PC or to the centrosome subcellular compartments, we next investigated the potential impact of the sponges on cortical interneuron migration in the case of no targeting to any specific subcellular compartment (Supplementary Fig. 2A). Of note, while SponGee and cAMP Sponge are expressed in the whole cytoplasm (Supplementary Fig. 2B–F), the cilioplasm remains devoid of any expression of the non-targeted sponges, most likely due to the transition zone acting as a physical barrier at its base (Supplementary Fig. 2B, C). No effect on cell directionality (Supplementary Fig. 2,G, H, J, K), migration speed (Supplementary Fig. 2I, L) or branching (Supplementary Fig. 2M–O) is observed in the absence of any subcellular targeting of the scavengers. This result was further validated by confirming the buffering efficiency of each cAMP (Supplementary Fig. 3A–F) and cGMP (Supplementary Fig. 3G–L) scavengers when expressed in the whole cytoplasm of migrating interneurons. Whereas soluble scavengers are excluded from the cilioplasm due to the presence of a filtering zone at the base of the PC, the centrosomal targeting of the cytoplasmic scavengers could be minimal, hence the absence of phenotype observed. Indeed, the

centrosome and its surrounding matrix form a highly organised and dense structure which is revealed by super-resolution fluorescence imaging[43], and are closely apposed to the Golgi apparatus[44], which could reduce the accessibility of soluble scavengers to the centrosome. Of note, our results do not exclude a cytoplamsic role for the cAMP and cGMP second messengers on cortical interneuron migration, since increasing cytoplasmic cAMP levels by using a pharmacological (forskolin bath application) or optogenetic approach (cytoplasmic light-sensitive adenylyl cyclase) strongly inhibits migration (Supplementary Fig. 4). Rather, our data suggest that a fine spatial and/or temporal dynamic remodelling of the cell can only be achieved by local and subtle variations of second messengers at the subcellular level, whether in the primary cilium, at the centrosome or within other cytoplasmic subcompartments.

## Increasing ciliary cAMP levels by photo-activation mimics the polarity defects induced by ciliary cGMP buffering

cGMP and cAMP signalling function together in many biological systems, and often oppose each other by regulating antagonistic functions. This could arise either as a consequence of their opposite regulation of common downstream effectors, or of their respective activation of distinct and opposing downstream signalling pathways[29,32,45]. An additional level of counteraction has been reported, by which the reciprocal cAMP/cGMP levels are negatively correlated to one another, with for example a cAMP increase associated to a cGMP reduction in several neuronal systems[30–32].

We thus wondered whether the opposite cell polarity phenotypes obtained by ciliary cGMP or cAMP buffering during migration could moreover rely on an opposite regulation of ciliary cAMP and cGMP levels. To address this question, we fused the light-sensitive adenylyl cyclase bPAC[37,46] to the 5HT6 sequence (Fig. 3A) and locally increased ciliary cAMP levels by photo-activation (Fig. 3B). Compared to control 5HT6-electroporated cells (Fig. 3C), 5HT6-bPAC electroporation and photo-activation (Fig. 3D) decreases the mean directionality ratios (Fig. 3E, F) and duration of persistent migration (Fig. 3G), as well as the average migration speed (Fig. 3H). Remarkably, these three migratory defects are those associated with ciliary cGMP buffering and changes in cell polarity (Fig. 1G, L–O), by opposition to the cell polarity maintenance phenotype induced by ciliary cAMP buffering (Fig.1H, P–R).

Our results are thus compatible with a ciliary balance model in which ciliary cAMP and cGMP levels oppose each other (Fig. 3I), since increasing ciliary cAMP levels by photo-activation mimics the phenotype obtained by ciliary cGMP buffering. In agreement with this model, inverting the ciliary cAMP/cGMP ratio by switching from cAMP photo-activation to cAMP buffering is sufficient to switch the cell polarity phenotype from frequent to rare changes in polarity (Fig. 3I–J, Supplementary Fig. 1).

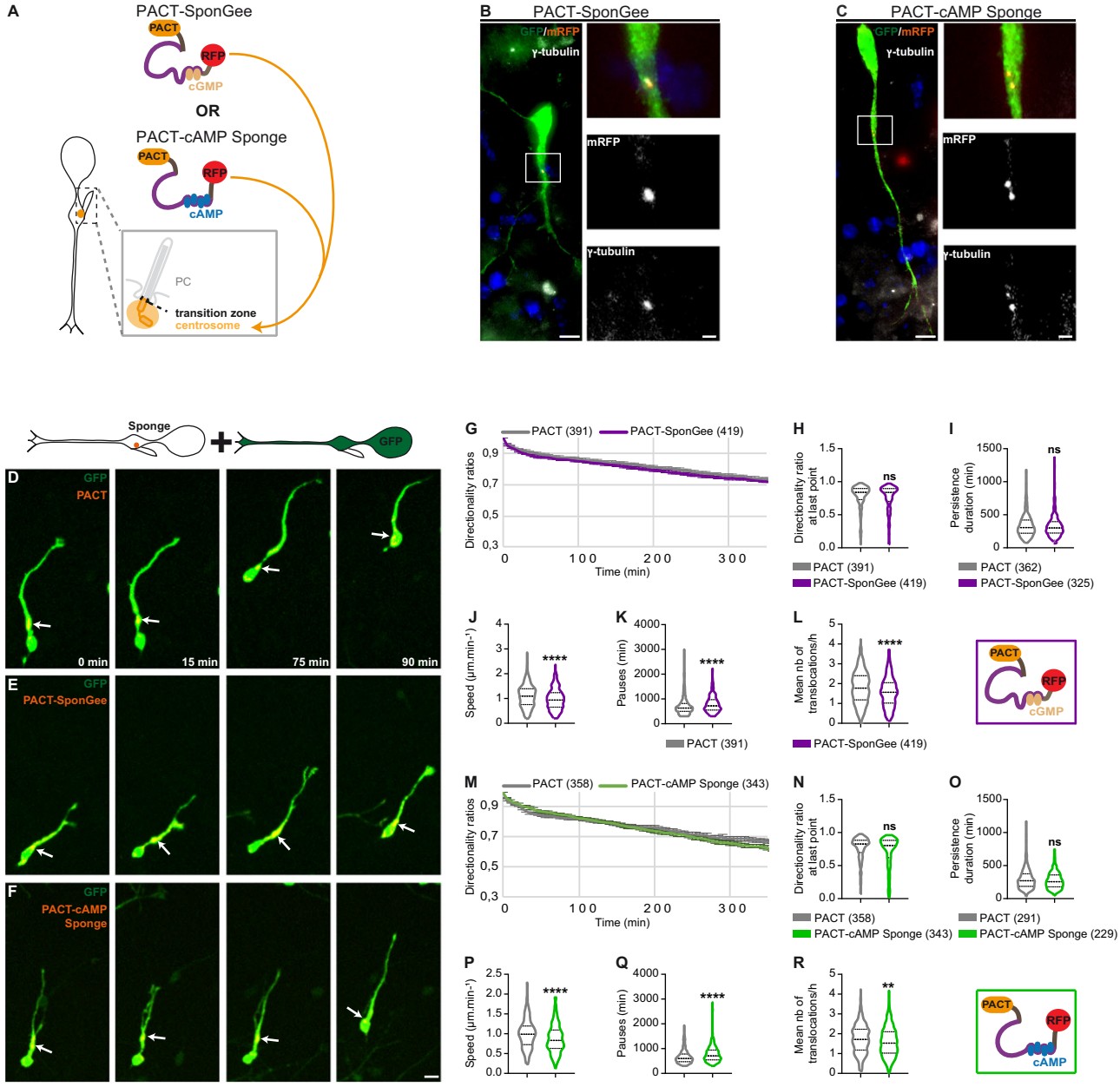

**Fig. 2 | Buffering cGMP or cAMP at the centrosome of migrating cortical interneurons similarly dysregulates nucleokinesis without affecting cell polarity. A** Representative scheme of the centrosome located in the cytoplasm at the base of the PC. The mRFP-tagged SponGee or cAMP Sponge scavengers are addressed to the centrosome by fusion to the PACT sequence. **B, C** High magnification of cortical interneurons co-electroporated with a cytoplasmic GFP construct and PACT-SponGee (**B**) or PACT-cAMP Sponge (**C**). Immunostaining with anti-GFP, anti-RFP and anti-γ-tubulin antibodies revealed the efficient co-localisation of the mRFP-tagged sponges with the centrosome. Insets are higher magnifications of the boxed region on the left. Scale bar, 5 μm; in insets, 1 μm. Co-localisation of our scavengers with γ-tubulin was observed in three independent experiments.

**D–F** Time-lapse recordings of cortical interneurons co-electroporated with the cytoplasmic GFP construct and the control mRFP-tagged PACT (**D**), PACT-SponGee (**E**) or PACT-cAMP Sponge (**F**). Arrows point at the mRFP-tagged centrosome. Scale bar, 10 μm. **G–R** Mean directionality ratios at each time point (**G, M**), after a maximum 350- (**H**) or 300-min migration period (**N**), mean persistence duration (**I, O**), mean migration speed (**J, P**) mean pause duration (**K, Q**) and mean number of translocations per hour (**L, R**) measured between the PACT and PACT-SponGee (**G–L**) or PACT-cAMP Sponge (**M–R**) conditions. The number of cells is indicated in the graph legends. **, $P \leq 0.01$, ****, $P \leq 0.0001$, ns, non significant. Two-tailed Mann–Whitney test (**H, I, J, K, L, N, O, P, Q, R**). Error bars are SEM. Source data and p values are provided as a Source data file.

## CXCL12 modulates the respective cAMP and cGMP levels in the PC to control cortical interneuron directionality

In vivo, cortical interneurons enter the cerebral cortex by its lateral border and migrate tangentially towards the medial cortex in the deep proliferative subventricular zone (SVZ) and in the superficial marginal zone (MZ). At any location along the tangential streams, a proportion of interneurons can re-orient radially in order to leave the tangential paths and to integrate the cortical plate (CP). This "tangential-to-radial

switch" of migration represents a major directional change operated by migrating cortical interneurons to reach their cortical target[47]. Among the guidance cues required for cortical interneurons to migrate in the embryonic cortex, the CXCL12 chemokine secreted by the MZ and by cortical progenitors in the SVZ/IZ (intermediate zone), promotes the cortical interneuron tangential migration within the deep and superficial tangential migratory streams through binding to the CXCR7 and CXCR4 receptors[22,23,48–53]. CXCL12 binding to CXCR4

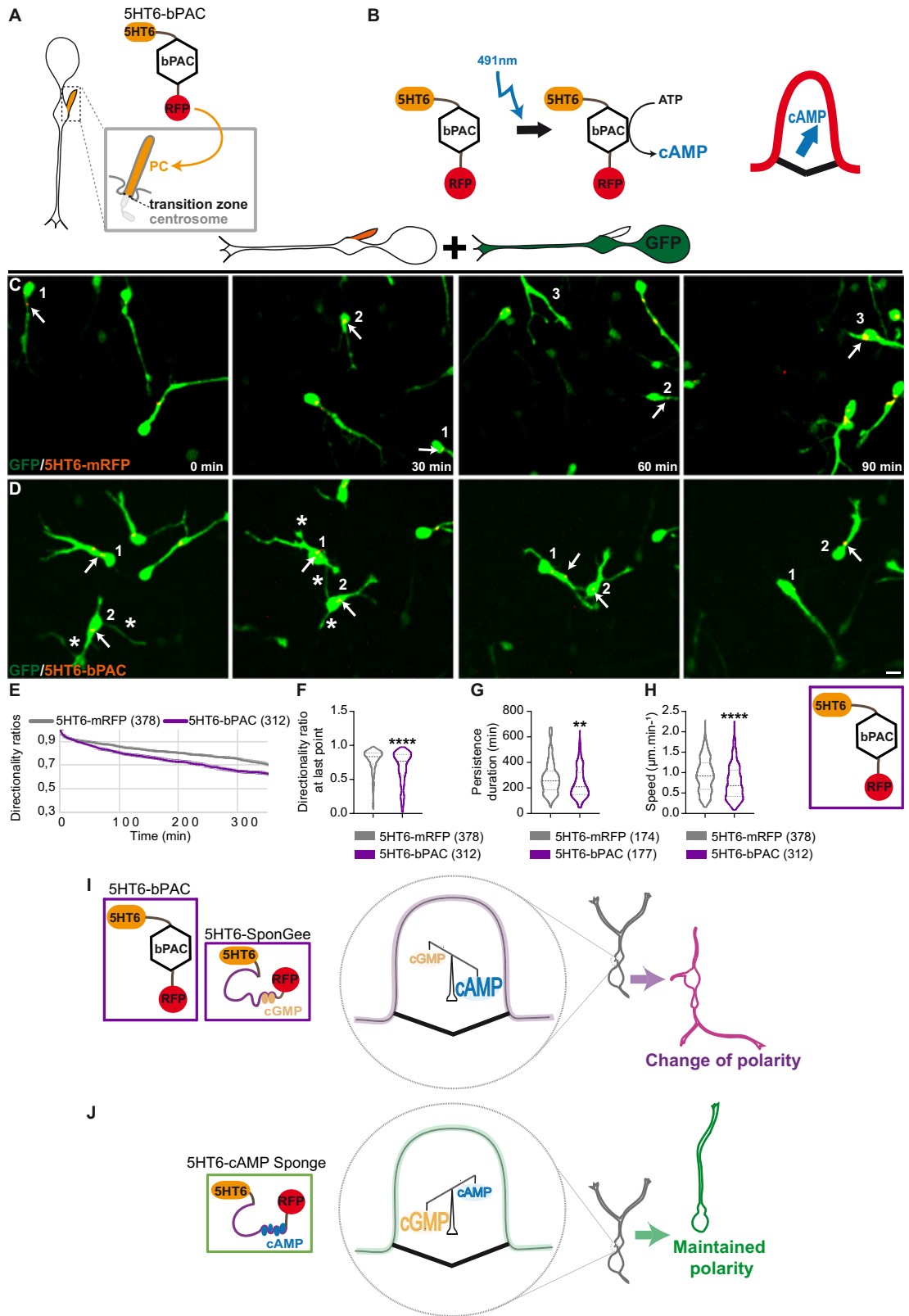

has moreover been reported to induce a Gi-mediated inhibition of adenylyl cyclase and reduction in cellular cAMP levels, leading to increased migration speed, reduced branching and subsequent increased migration directionality within the tangential stream[22]. Moreover, the CXCR4 receptor for CXCL12 has been found in cortical interneuron primary cilia[7], in addition to its localisation at the leading process and soma[23].

Here, we examined whether the CXCL12 effect on increased cortical interneuron directionality may be the result of a local CXCL12 effect at the PC. To test our hypothesis, we first took advantage of our in vitro co-culture model to assess the influence of CXCL12 on the directionality and migration speed of cortical interneurons electroporated with constructs that specifically erase cAMP or cGMP signals in the PC. Confocal time lapse imaging was initiated as cells started their

**Fig. 3 | Increasing ciliary cAMP levels by photo-activation induces frequent changes in polarity. A** Representative scheme of the cortical interneuron PC, anchored to the centrosome via the mother centriole and physically separated from the cytoplasm by the transition zone. The mRFP-tagged bPAC construct is targeted to the PC by fusion to the 5HT6 targeting sequence. **B** 5HT6-bPAC is photo-activated by blue light (491 nm laser) every minute for 2,1 s, leading to increased ciliary cAMP production. **C, D** Time-lapse recordings of cortical interneurons co-electroporated with the GFP cytoplasmic construct and the control mRFP-tagged 5HT6 (**C**) or 5HT6-bPAC (**D**). Arrows and asterisks point at the dynamic mRFP-tagged PC and branch formation at the soma, respectively. Scale bar, 10 µm. **E** Graphical representation of the mean directionality ratios at each time point. **F** Mean directionality ratio after a maximum 350-minute migration period. **G, H** Mean persistence duration (**G**) and migration speed (**H**). **I, J** Model depicting a ciliary cAMP/cGMP balance mechanism that regulates cortical interneuron polarity. Electroporation with 5HT6-bPAC or 5HT6-SponGee favours a ciliary balance conformation with higher cAMP levels compared to cGMP, and is associated with frequent changes of polarity (**I**). By contrast, inverting this ciliary cAMP/cGMP balance by electroporation of 5HT6-cAMP Sponge inverts the polarity phenotype from frequent to rare changes of polarity (and vice versa; **J**). The number of cells is indicated in the graph legends. **, $P \le 0.01$, ****, $P \le 0.0001$, ns, non significant. Two-tailed Mann–Whitney test (**F–H**). Error bars are SEM. Source data and $p$ values are provided as a Source data file.

migration. CXCL12 was added to the culture medium after 5 h and imaging continued for another 10 h (Fig. 4A–C). Importantly, analysis of the migratory behaviours of electroporated MGE cells prior to drug application reproduces the phenotypes obtained previously (Fig. 4B, D): while ciliary cAMP buffering increases directionality ratios (Fig. 4D, F; light grey and light green curves and violin plots) without affecting the average migration speed (Fig. 4G; light grey and light green violin plots), ciliary cGMP buffering reduces both directionality ratios (Fig. 4D, F; light grey and light purple curves and violin plots) and migration speed compared to controls (Fig. 4G; light grey and light purple violin plots). CXCL12 application on control 5HT6-electroporated cells (Fig. 4C, left sequence) increases directionality ratios (Fig. 4E, F; light and dark grey curves and violin plots) to the same levels as those observed prior to drug application for MGE cells electroporated with 5HT6-cAMP Sponge (Fig. 4F; dark grey and light green violin plots). The mean migration speed of control cells is also increased by CXCL12 application (Fig. 4G, light and dark grey violin plots), as previously observed[22]. CXCL12 therefore appears sufficient to convert the directionality of a "control-like" migrating cell to a "5HT6-cAMP Sponge-like" migratory behaviour. In agreement, CXCL12 application on cells electroporated with 5HT6-cAMP Sponge (Fig. 4C, middle sequence) maintains directionality ratios at the same high levels as observed prior to drug application (Fig. 4F; light and dark green violin plots). On the other hand, CXCL12 application on cells electroporated with 5HT6-SponGee (Fig. 4C, right sequence), which phenocopies cAMP production in the PC, is sufficient to switch directionality ratios from low – prior to drug application – to high (Fig. 4D–F, light and dark purple curves and violin plots), reaching levels comparable to those induced by CXCL12 application on controls (Fig. 4E–F, dark purple and dark grey curves and violin plots) or to 5HT6-cAMP Sponge expression (Fig. 4D–F; dark purple and light green curves and violin plots). CXCL12 application on 5HT6-SponGee-electroporated cells is therefore sufficient to convert the low directionality and speed phenotype of cells displaying a low ciliary cGMP/cAMP ratio (Fig. 4D, 5HT6-Sponge-electroporated cells, light purple curve) to the highly directional phenotype of cells displaying a high ciliary cGMP/cAMP ratio (Fig. 4D, E, 5HT6-cAMP Sponge-electroporated cells, light and dark green curves).

Importantly, bath application of the CXCR4 antagonist AMD3100 in addition to CXCL12 (Fig. 5A) was sufficient to abolish this conversion and maintain the low directionality and speed phenotypes of 5HT6-SponGee-electroporated cells (Fig. 5B–J). To further ascertain that the CXCL12-induced switch from low to high migration directionality is indeed the result of a switch from a low ciliary cGMP/cAMP ratio to a high ciliary cGMP/cAMP ratio mediated by a ciliary CXCL12/CXCR4 inhibition of cAMP, we next sought to block the ciliary cGMP/cAMP ratio in a low configuration. This was achieved by targeting and photo-activating the optogenetic adenylyl cyclase bPAC specifically to the primary cilium (Fig. 5K–L). Indeed, while CXCL12 induces a Gi-mediated inhibition of transmembrane adenylyl cyclases, this inhibition cannot include bPAC, which is a bacterial soluble adenylyl cyclase. bPAC activity is tightly correlated with the state of its light-sensitive

domain[46] and is thus not affected by mammalian G proteins. This experimental setup therefore allows cAMP levels to drop in response to CXCL12 at the whole cell surface – where CXCR4 is expressed –, at the exclusion of the ciliary compartment where the effect of CXCL12 is counteracted by 5HT6-bPAC photo-activation. Our results show that blocking the CXCL12-mediated decrease of cAMP levels specifically within the primary cilium is sufficient to abolish the cells' response to CXCL12 – i.e., the switch from low to high directionality and speed (Fig. 5M–O). Of note, immunohistochemistry experiments in the presence of forskolin alone or in combination with CXCL12 confirmed that CXCL12 application decreases the punctate cAMP immunostaining pattern – which has already been observed in prior studies[54,55] – detected within the primary cilium (Supplementary Fig. 5). Further confirming the cell's requirement of the primary cilium compartment in mediating the CXCL12-induced increase in migration directionality, migrating cortical interneurons no longer respond to CXCL12 following a genetic ablation of the primary cilium (Supplementary Fig. 6).

Taken together, our results identify the primary cilium compartment, and more specifically the ciliary cGMP/cAMP ratio, as a target of the CXCL12/CXCR4 signalling pathway required to regulate cortical interneuron polarity during migration (Supplementary Fig. 7).

## The ciliary cGMP/cAMP ratio regulates cortical interneuron directionality ex vivo within the deep tangential stream

Finally, we examined whether scavengers targeted to the PC (Fig. 6A) can control the tangential to radial switch of migrating interneurons within the cortical structure. Electroporated MGE explants were grafted at the subpallium/pallium boundary of E15.5 brain organotypic slices, a stage at which the embryonic cortex exhibits visible SVZ, ZI and CP layers (Fig. 6B). After 12 to 16 h culture, MGE-derived cortical interneurons exit the explant and start to migrate within the structured cortical substrate. Cells were left to migrate for an extra 48 h before fixation. Due to the graft position in the host cortex, control cortical interneurons (Fig. 6C) exiting the grafted MGE start to migrate in a highly directional tangential stream within the VZ/SVZ regions boarding the ventricle, where CXCL12 is expressed[50,53,56]. During their tangential progression, interneurons re-orient themselves in the SVZ/IZ by extending a radially oriented leading process towards the developing cortical plate. To assess the physiological role of PC-elicited second messenger signals on directional changes, we evaluated the tangential-to-radial orientation switch of MGE cells electroporated with PC-targeted scavengers within the SVZ of grafted cortical slices (Fig. 6D-E). The leading process angular orientation was measured as depicted in Fig. 6F and angles were categorised as low/tangential, intermediate or high/radial. Remarkably, while ciliary cAMP buffering increases the proportion of tangentially-oriented cells over the radial ones, ciliary cGMP buffering favours an increased leading process radial orientation, reflecting an increased radial migration switch towards the developing cortical plate (Fig. 6G). Therefore, locally buffering cGMP or cAMP levels within the PC of ex vivo migrating cortical interneurons is sufficient to alter the tangential-to-

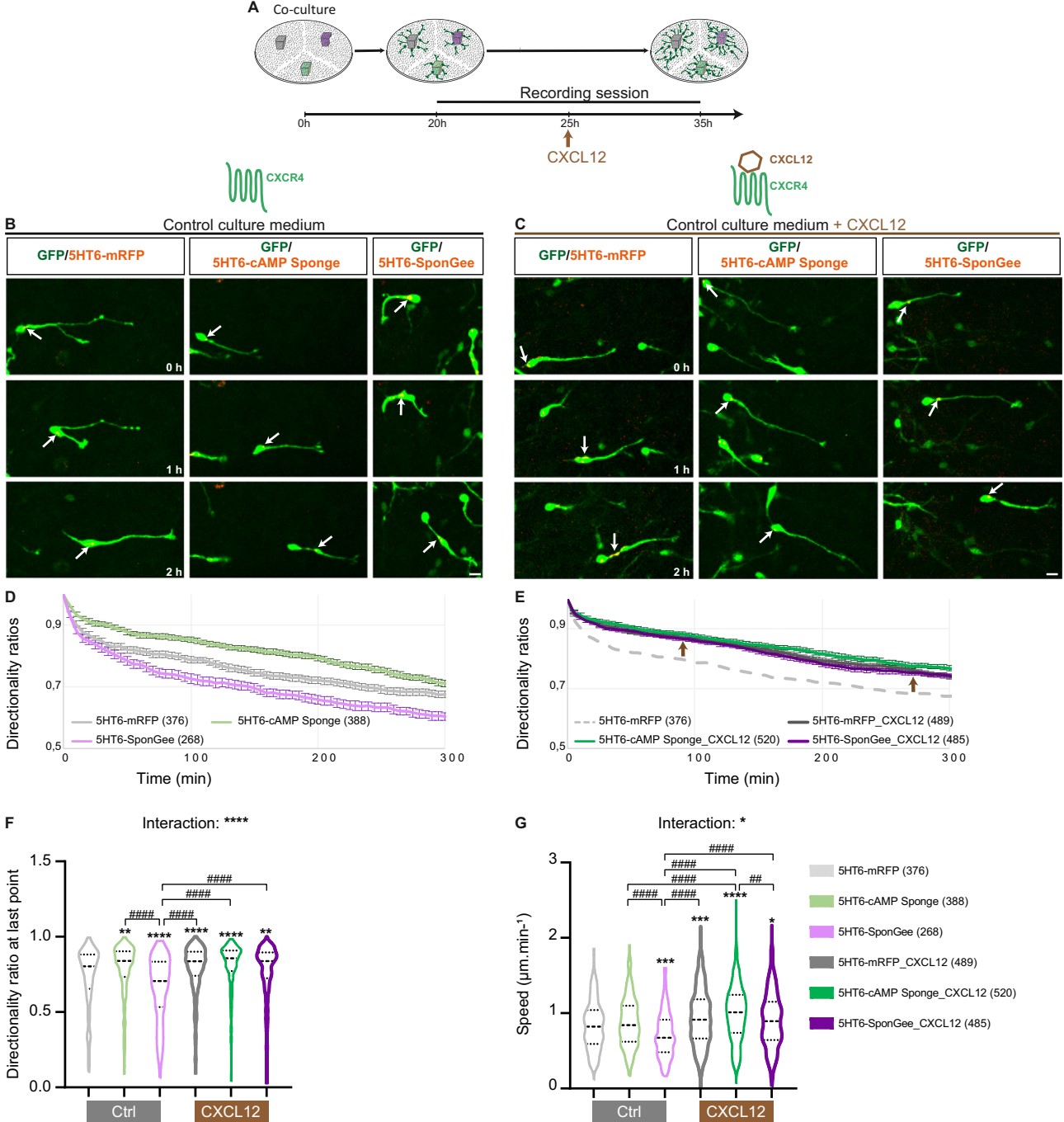

**Fig. 4 | CXCL12 bath application increases cell directionality in a way that mimics ciliary cAMP buffering. A** Representative scheme of the in vitro protocol. MGE explants electroporated with 5HT6 (grey), 5HT6-SponGee (purple) and 5HT6-cAMP Sponge (green) are co-cultured on dissociated cortical cells. Live imaging starts as cortical interneurons initiate their migration and continues for 10 h after CXCL12 is added to the culture medium. **B, C** Time-lapse recordings of cortical interneurons migrating in control medium (**B**) or after CXCL12 bath application (**C**). Interneurons were co-electroporated with the GFP cytoplasmic construct and the control mRFP-tagged 5HT6 (**B, C**, left-hand sequence), 5HT6-cAMP Sponge (**B, C**, middle) or 5HT6-SponGee (**B, C**, right-hand sequence). Arrows point at the dynamic mRFP-tagged PC. Scale bar, 10 μm. **D, E** Graphical representation of the mean directionality ratios at each time point over a 5-h period, prior to (light curves, **D**) and after (dark curves, **E**) CXCL12 application, for the 5HT6, 5HT6-cAMP Sponge and 5HT6-SponGee conditions. Directionality ratios are represented on two sets of graphs for more clarity, although all

directionality ratios (prior or after drug application) were measured during a same experimental setup. The light dotted grey curve depicted in (**E**) corresponds to the control 5HT6-mRFP condition prior to drug application that is also represented in (**D**). Brown arrows highlight the increased directionality induced by CXCL12 application, independently of the electroporated construct. **F, G** Mean directionality ratio after a maximum 5-hour migration period (**F**) and mean migration speed (**G**) for the 5HT6, 5HT6-cAMP Sponge and 5HT6-SponGee conditions prior to and after CXCL12 bath application. The number of cells is indicated in the graph legends. Statistically significant differences are reported using the * or # symbols, when comparing values to the 5HT6 condition or between non-control conditions, respectively. *, $P \leq 0.05$; **, $P \leq 0.01$; ***, $P \leq 0.001$; **** or ####, $P \leq 0.0001$. Two-way ANOVA test with Bonferroni's multiple comparison post test. **F** Interaction: ****; Genotype effect:****; Treatment effect: ****. **G** Interaction: *; Genotype effect: ****; Treatment effect: ****. Error bars are SEM. Source data and p values are provided as a Source data file.

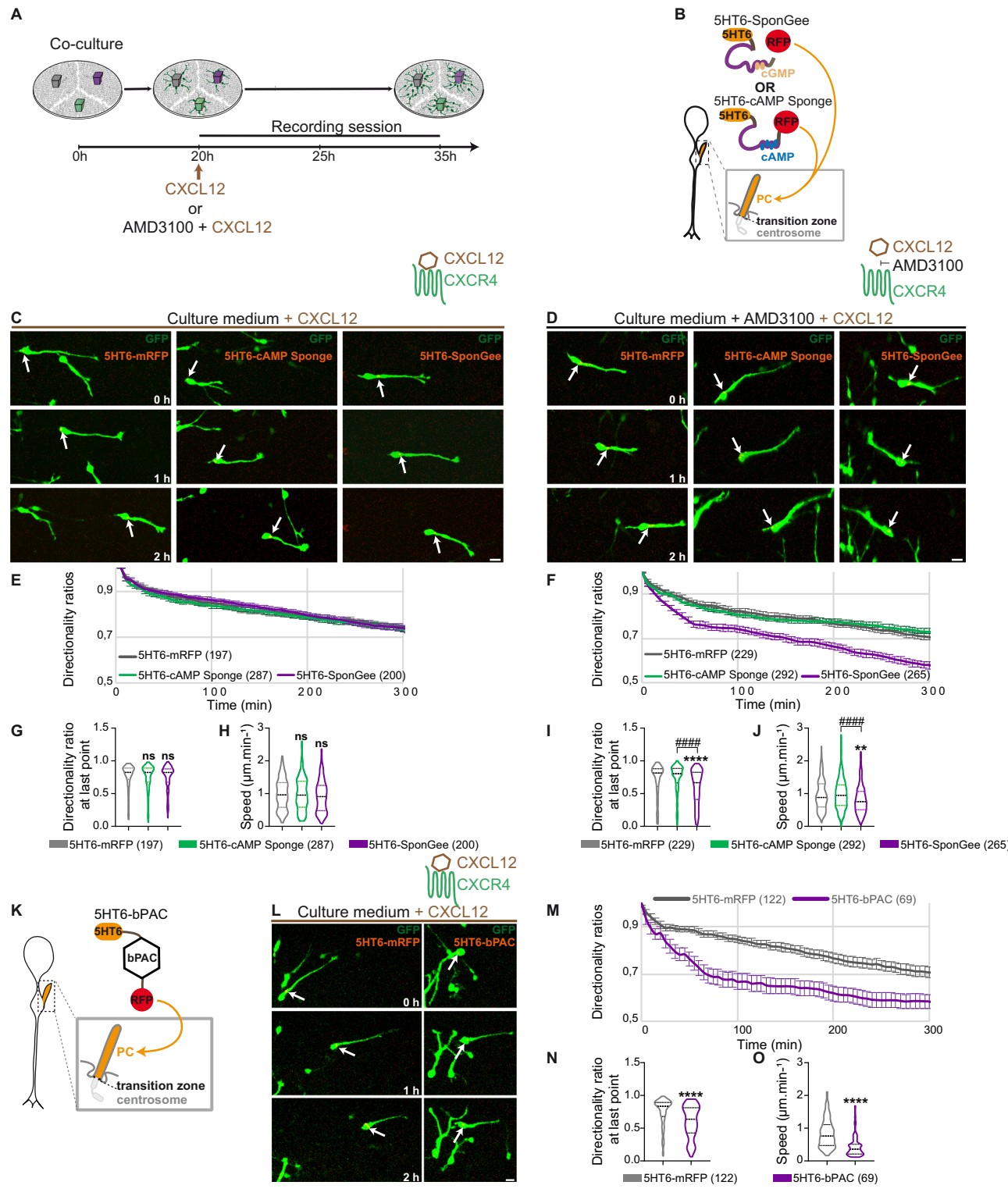

radial migration switch occurring within the SVZ. Taken together, our results moreover highlight a strong coherence between the in vitro and ex vivo data. Indeed, both in vitro and ex vivo buffering of ciliary cGMP are responsible for polarity-driven changes in directionality, likely operated from the cell body compartment, while ciliary cAMP buffering maintains cell directionality and favours tangential migration.

Altogether, our results support a model in which CXCL12 secreted by SVZ cortical progenitors binds to ciliary-located CXCR4 receptors on tangentially migrating MGE cells, thereby reducing ciliary cAMP

levels. This stabilises the ciliary cAMP/cGMP balance in a conformation that induces a cell polarity maintenance phenotype and a highly directional tangential migration mode (Fig. 7).

## Discussion

Our study sheds light on a signalling mechanism controlling the directionality of cortical interneuron migration that originates at the PC and involves a negative crosstalk between cGMP and cAMP signals. This identified role for the PC as the steering wh`eel of cortical interneuron migration is independent of the

**Fig. 5 | CXCL12 influence on cell polarity requires its binding to CXCR4 and a decrease in ciliary cAMP levels. A** MGE explants electroporated with 5HT6 (grey), 5HT6-SponGee (purple) and 5HT6-cAMP Sponge (green) are co-cultured on dissociated cortical cells. After 20 h, CXCL12, with or without AMD3100, is added to the culture medium. Live imaging is then performed for 15 h. **B** The mRFP-tagged SponGee or cAMP Sponge are fused to 5HT6 for PC targeting. **C, D** Time-lapse recordings of cortical interneurons co-electroporated with the GFP cytoplasmic construct and the PC-targeted 5HT6-mRFP (**C, D**, left-hand sequences), 5HT6-cAMP Sponge (**C, D**, middle sequences) or 5HT6-SponGee (**C, D**, right-hand sequences). Imaging was performed in the presence of CXCL12 (**C**) or CXCL12 and AMD3100 (**D**). White arrows indicate the dynamic mRFP-tagged PC. Scale bar, 10 µm. **E, F** Mean directionality ratios at each time point over a 5-h period in the presence of CXCL12 (**E**) or AMD3100 + CXCL12 (**F**). **G–J** Mean directionality ratio after a maximum 5-h migration period (**G, I**) and mean migration speed (**H, J**) for 5HT6-mRFP-, 5HT6-cAMP Sponge and 5HT6-SponGee-electroporated cells in the presence of CXCL12 (**G, H**) or AMD3100 + CXCL12 (**I, J**). **K** The mRFP-tagged bPAC construct is fused to 5HT6 for PC targeting. **L** Time-lapse recordings of cortical interneurons co-electroporated with the GFP cytoplasmic construct and the PC-targeted 5HT6-mRFP (left) and bPAC (right) constructs. Imaging was performed in the presence of CXCL12. Scale bar, 10 µm. **M** Mean directionality ratios at each time point over a 5-h period. **N** Mean directionality ratio after a maximum 5-h migration period. **O** Mean migration speed. The number of cells is indicated in the graph legends. Statistically significant differences are reported using the * or # symbols, when comparing values to the 5HT6 condition or between non-control conditions, respectively. **, $P \leq 0.01$; $P \leq 0.0001$; **** or ####, $P \leq 0.0001$, ns, non significant. Kruskal-Wallis test (**G, H; I, J**) with Dunn's multiple comparisons test (**I, J**) and two-tailed Mann–Whitney test (**N–O**). Error bars are SEM. Source data and p values are provided as a Source data file.

cGMP- and cAMP-dependent cell motility process which is concomitantly regulated by the centrosome compartment. Finally, our results directly link the CXCL12/CXCR4 signalling pathway to PC-elicited second messengers, resulting in the precise regulation of the tangential-to-radial migration switch of cortical interneurons.

### The PC signalling hub regulates cortical interneuron polarity via a ciliary cAMP/cGMP balance mechanism that opposes cAMP and cGMP downstream signalling

The PC is an evolutionary conserved organelle specialised in cAMP and cGMP signalling[15]. The strength of our approach relies on the specific manipulation of PC-located second messenger signals by combining the use of highly specific scavengers[36–38,57] and photo-activated constructs to their efficient targeting to the PC compartment via the 5HT6 sequence. Further validating our approach, compared to 5HT6 electroporation, electroporation of MGE cells with the 5HT6 receptor harbouring the Gs-dead mutation – known to abolish both the 5HT6 constitutive activity and the increased primary cilium length induced by 5HT6 overexpression[40,58] – does not affect cell directionality or migration speed. The negative crosstalk we report between ciliary cGMP and cAMP signals is in line with other neuronal studies[29–32]. Although the mechanisms responsible for this opposition remain unknown, a reciprocal inhibition between each second messenger through phosphodiesterase-dependent hydrolysis has been proposed[32,45,59]. Our data are therefore compatible with a ciliary cAMP/cGMP balance mechanism in which high ciliary cAMP levels lead to PDE-dependent cGMP degradation, thereby inducing the same change in polarity phenotype as ciliary cGMP buffering (Fig. 3).

A second level opposing ciliary cGMP and cAMP may arise from their respective activation of two distinct and opposing downstream signalling pathways, since buffering ciliary cGMP or cAMP impacts two distinct cellular compartments. While reduced ciliary cGMP signals may trigger a pathway that locally increases branching at the soma/swelling, reduced ciliary cAMP levels may lead to a signalling cascade reaching down to the leading process to inhibit branch formation and induce directional cell migration (Fig. 1). In both cases, the cytoskeleton is a likely downstream effector, especially since studies have already pointed it out as a downstream target of ciliary signals[60–62].

### Cortical interneurons spatially organise additional cAMP and cGMP signalling compartments outside of the PC to regulate other aspects of migration

The ciliary cAMP/cGMP negative crosstalk regulating cell polarity is specific to the PC compartment, both functionally and mechanistically. Functionally, cGMP or cAMP buffering at the centrosome alters nucleokinesis, but not cell polarity. Moreover, unlike at the PC, we do not observe an opposition between cAMP and cGMP signals at the centrosome, since the buffering of either second messenger leads to the same dysregulation of nucleokinesis (Fig. 2). These results are in

favour of a positive cAMP/cGMP crosstalk at the centrosome, with cAMP and cGMP signals converging on a pool of downstream effectors involved in a same process, rather than opposing each other. Such a positive crosstalk has already been observed in platelets[27,28], T-cells[63] or olfactory sensory neurons[64]. Our results are thus in line with several studies reporting the spatial segregation of cAMP and cGMP signals within different cytoplasmic microdomains[12,29,34,35] and provide a concrete example of how cells may spatially organise such signals as well as their interplay to regulate different aspects of neuronal migration. Remarkably, another such example has also been evidenced during RGC axon guidance, in the lipid raft and non-lipid raft subcellular compartments[57]. We here moreover confirm previous data reporting a deficient nucleokinesis associated with cAMP delocalisation from the centrosome[65] (Fig. 2). Interestingly, although the centrosome had previously been described as a cAMP signalling centre during cell cycle progression or neuronal migration[65,66], its role in cGMP signalling had so far – to our knowledge – not been documented. Our study therefore brings new insight into the cAMP/cGMP interplay operating at this subcellular compartment.

### The extracellular CXCL12 chemokine targets ciliary second messengers and acts as a switch to set a directional migration mode in the deep tangential stream

During development, CXCL12 mediates the directional tangential migration of cortical interneurons within the deep migratory stream, in a process involving a CXCR4-mediated decrease in cAMP levels[22,48]. Our results strongly suggest that the cAMP-dependent regulation of cell branching and directionality originates from a reduced ciliary cAMP activity, since ciliary cAMP buffering is sufficient to reproduce the CXCL12-mediated increase of cell directionality and reduced branching at the leading process (Fig. 1 and Fig. 4). This highlights the role of PC cAMP/cGMP signalling as the steering wheel of migration. We propose a model in which CXCL12 induces a CXCR4-mediated "ciliary cAMP buffered-like" balance conformation leading to a highly directional migration mode in the SVZ (Fig. 7). Importantly, CXCL12 secretion by progenitor cells has been reported to decrease over time in a temporal pattern coinciding with the radial re-orientation of cortical interneurons towards the developing CP[67]. Conceptually, the ciliary cAMP/cGMP balance conformation switch that induces polarity changes and the radial re-orientation of migrating cells (Fig. 6) may therefore be triggered by switching off the CXCL12 signal: as CXCL12 expression decreases, ciliary cAMP levels may be allowed to increase, leading to a gradual inhibition of ciliary cGMP levels through PDE-mediated hydrolysis of ciliary cGMP[32]. We propose that switching off CXCL12 may lead to a ciliary cAMP/cGMP balance shift favouring new branch formation towards the developing CP. This may in turn enable cells to respond to other extracellular molecules of the environment, such as the Shh morphogen, previously described to promote the tangential to radial switch of migrating cortical interneurons[6] – in a yet undescribed mechanism.

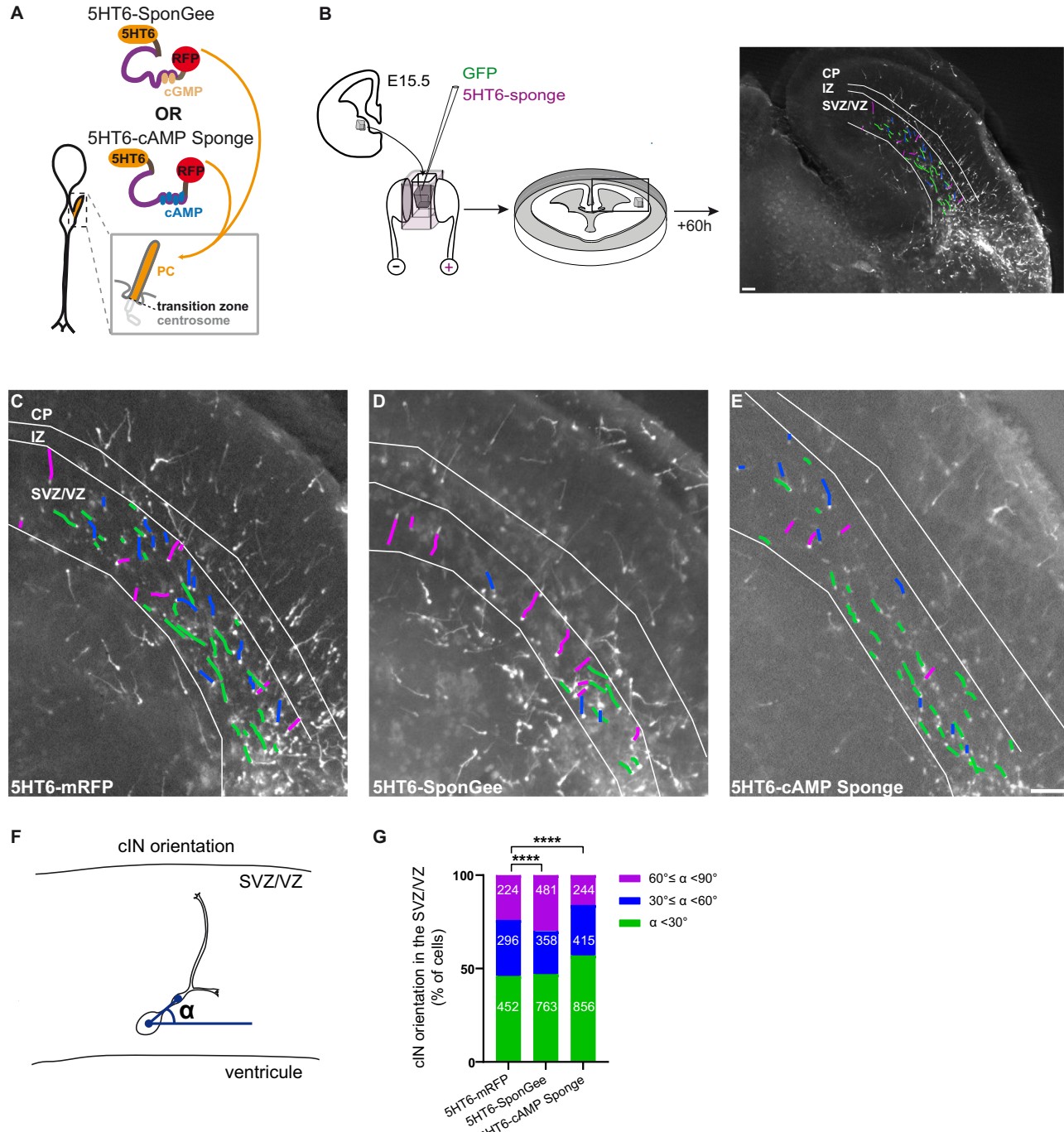

**Fig. 6 | Ciliary cGMP or cAMP buffering impairs the orientation of cortical interneurons migrating ex vivo in the SVZ. A** Representative scheme of a cortical interneuron PC. The mRFP-tagged SponGee or cAMP Sponge chelators are fused to the 5HT6 sequence for PC targeting. **B** MGEs are dissected from E15.5 mouse embryos and co-electroporated with the cytoplasmic GFP construct and the desired scavenger. Electroporated MGEs are then grafted at the pallium-subpallium boundary of E15.5 organotypic slices and cells are left to migrate for 60 h before fixation. Scale bar, 100 μm. **C–E** Epifluorescence acquisition of E15.5 brain organotypic slices grafted with MGEs co-electroporated with GFP and 5HT6 (**C**), 5HT6-SponGee (**D**) or 5HT6-cAMP Sponge (**E**). Slices were immunostained with the anti-RFP and anti-GFP antibodies. For clarity's sake, only the GFP staining is represented. SVZ/VZ, IZ and CP regions were delimited using the dapi staining. Leading processes of migrating cells are drawn in green, purple or blue according to the value of their orientation angle (see **F**, **G**, below), namely tangential to the ventricle, radial or intermediate (respectively). Scale bar, 100 μm. **F** Representative scheme of the orientation angle (α) measured for each migrating cortical interneuron between the soma-swelling axis and the tangential to the ventricle. **G** α orientation angles were distributed in three categories corresponding to low (<30° or tangential orientation; green), high (>60° or radial orientation; purple) and intermediate angles (30° ≤ α < 60°; blue). The number of cells is indicated on graphs. Orientation angles were measured for cells obtained from four (5HT6-mRFP, 16 organotypic slices), five (5HT6-SponGee, 19 organotypic slices) and five (5HT6-cAMP Sponge, 16 organotypic slices) embryonic brains. Differential distribution of these angles was tested for 5HT6-cAMP Sponge- and 5HT6-SponGee-electroporated cells compared to controls using two-tailed Chi-square tests. ****, $P \leq 0.0001$. SVZ subventricular zone, VZ ventricular zone, IZ intermediate zone, CP cortical plate, cIN cortical interneuron. Source data and p values are provided as a Source data file.

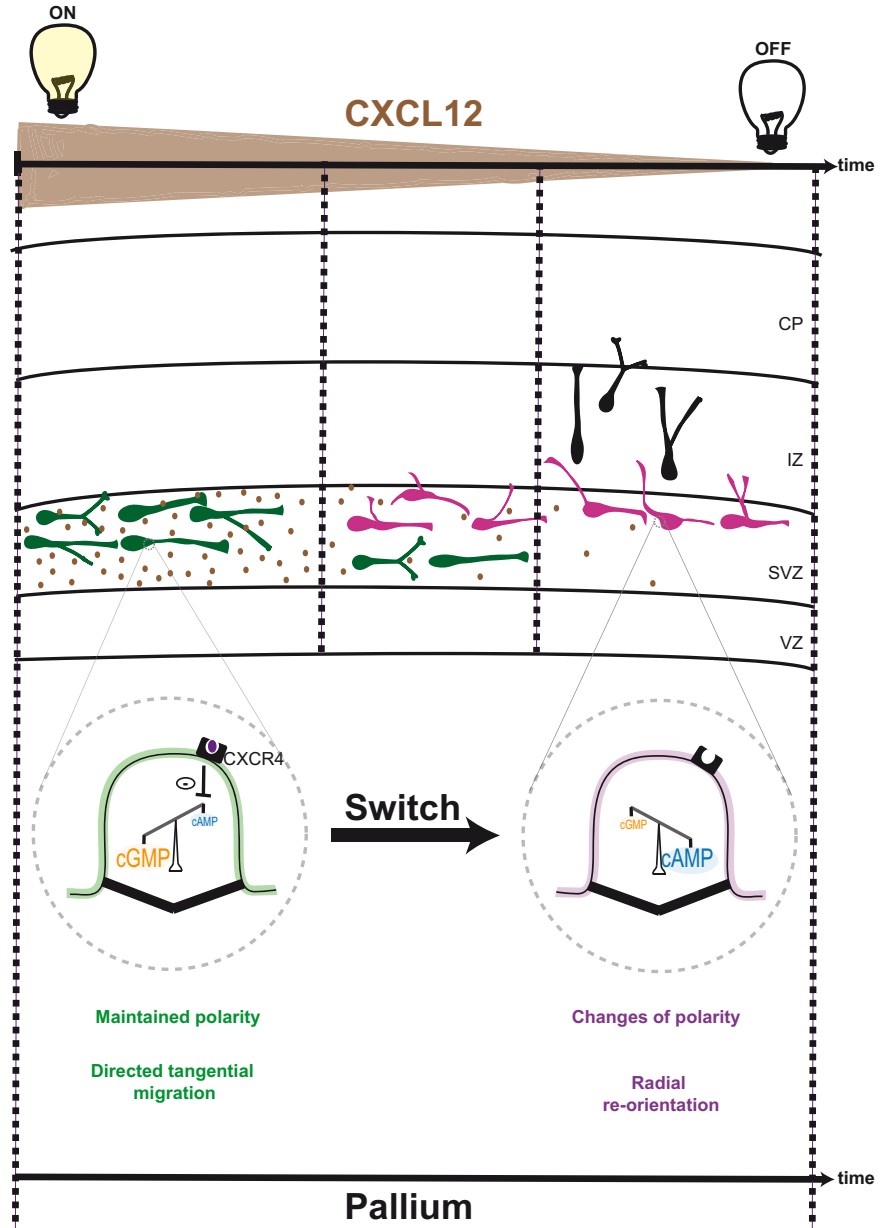

**Fig. 7 | Model of the ciliary cAMP/cGMP switch activated by CXCL12 to set the tangential migration mode of cortical interneurons in SVZ.** Within the SVZ, ventrally-born cortical interneurons (in green) are exposed to CXCL12 secreted by cortical intermediate progenitors. CXCL12 binds to the ciliary CXCR4 receptor of migrating cells, thereby inhibiting adenylyl cyclase-dependent cAMP production within the PC. The ciliary cAMP/cGMP balance is stabilised in a conformation with low ciliary cAMP levels compared to cGMP (Fig. 3, magnified PC in green), stabilising the polarity of migrating cells within the deep tangential stream, promoting sustained tangential migration. The proportion of tangentially oriented cells therefore increases compared to control cells, at the expense of radially oriented cells (Fig. 6). As CXCL12 expression decreases along time, adenylyl cyclase-dependent production of ciliary cAMP resumes gradually and ciliary cAMP levels increase, favouring a ciliary balance conformation with high cAMP levels compared to cGMP (magnified PC in purple). Cortical interneuron polarity is consequently unlocked and cortical interneurons extend new processes from the soma-swelling compartment, which can become new leading processes. The proportion of cells with radially oriented leading processes increases compared to controls (Fig. 6). CXCL12 expression is represented in brown as a decreasing gradient on the time axis. VZ ventricular zone, SVZ subventricular zone, IZ intermediate zone, CP cortical plate.

## Methods

### Mice

Mouse embryos were collected from adult pregnant Swiss mice ordered from Janvier or from *Kif3a^{fl/fl}* mice. Experiments were performed in the lab (Institut du Fer à Moulin, approval number D 75-05-22) in a conventional animal facility according to European guidelines. Experiments performed in the present study have been validated and approved by the Ethical committee Charles Darwin (C2EA-05, authorized projects 02241.02 and 29407).

### Molecular biology

**Control constructs.** The pCX-5HT6-mCherry-GFP plasmid was a kind gift from the Nicol lab.

The pCAGGS-PACT-mKO1 and pCAGGS-PACT-GFP plasmids were kind gifts from Pr. Fumio Matsuzaki[42]. pCAGGS-eGFP and pCAGGS-Cre were kindly gifted by Pr. Fujio Murakami.

Mut5HT6-mRFP was generated by digestion of pCX-5HT6-mCherry-GFP with AgeI.

The mutated F69LT70ID72A 5HT6 sequence was generated using two rounds of PCR amplification with the Phusion DNA polymerase.

First, we amplified the N-Ter moiety with 5HT6_for (5′- GGC AAA-GAATTCTGATATCTTTAATCGCCACCATGGTT −3′) and mut_5HT6_rev (5′-CACCAATCCCACC  ATCAGGGCCGATATTAAGAGCGACACCAGGA AGAAG-3′), and the C-ter moiety using mut_5HT6_for (5′- CTTC TTCCTGGTGTCGCTCTTAATATCGGCCCTGATGGTGGGATTGGT-3′) and 5HT6_rev (5′-GGAGCTAGCAACCGGTCCTCCTGC-3′).

The purified N-ter end C-ter moieties were diluted, mixed and amplified with the Phusion DNA polymerase using 5HT6_mcherry_for (5′-TAATTAAACCCCGGGACCGGTGGATCCGG-3′) and 5HT6_mCher-ry_rev (5′-CATGGTGGCGACCGGTCCTCCTG-3′).

The full length mutated 5HT6 sequence was cloned using the infusion cloning system (Takara).

**Non-targeted scavengers.** The pCX-SponGee-mRFP, pCX-cAMP Sponge-mRFP and pcx-mRFP-bPAC plasmids were kind gifts from the Nicol lab.

**Primary cilium-targeted constructs.** pCX-5HT6-SponGee-mRFP was generated by digesting pCX-SponGee-mRFP with the restriction enzymes PacI and AgeI. 5HT6 was PCR-amplified from the pCX-5HT6-mCherry-GFP plasmid, using the CloneAmp HiFi PCR Premix (Ozyme, France); forward primer (5′-GGCAAAGAATTCTGATATCT TTAATCGC-CACCATGGTT-3′); reverse primer (5′-GCTCCGGAGCTAGCAACCG GTCCTCCT-3′).

pCX-5HT6-cAMP Sponge-mRFP was generated by digesting pCX-5HT6-SpiCee-mRFP[68] with the restriction enzymes BmtI and BglII. cAMP Sponge-mRFP was PCR-amplified from the pCDNA3-cAMP Sponge-mRFP plasmid, using the CloneAmp HiFi PCR Premix (Ozyme, France); forward primer (5′-AGGACCGGTTGCTAGGGC-CATCTCCAAGA-3′); reverse primer (5′-TTTTGGCA GAGGGAAAAA-GATCTAGGCGCCGGTGG-3′).

Both plasmids were cloned using the NEBuilder HiFi DNA Assembly Cloning kit (New England Biolabs, UK).

pCX-5HT6-mRFP-bPAC was generated by digestion of pCX-LynLyn-mRFP (kind gift from the Nicol lab) with EcoRV and NotI. 5HT6 was PCR amplified from pCX-5HT6-SpiCee-mRFP-eGFP with the Phusion DNA Polymerase (NEB; forward primer: 5′-GCAAA-GAATTCTGATGCCACCATGG TTCCAGAGC-3′; reverse primer: 5′-GAG GAGGCTGCGGCCGCACTGTTCATGGGGGAACCAAG-3′).

The PCR fragment was cloned upstream of mRFP into pCX-LynLyn-mRFP-bPAC using the infusion cloning system (Takara)

**Centrosome-targeted scavengers.** pCX-PACT-SponGee was generated by digestion of pCX-5HT6-Spongee by AgeI and EcoRV.

pCX-PACT-cAMP Sponge was generated by digestion of pCX-5HT6-cAMP Sponge-mRFP by AgeI and EcoRV.

The 674 bp PACT domain of mouse pericentrin, including a Kozak sequence upstream of the ATG initiation codon, was amplified from the pCAGGS-PACT-GFP plasmid (forward primer: 5′-GCAAA-GAATTCTGATGCCACCATGGACCCAGAGTGGC-3′; reverse primer: 5′-GGAGCTAGCAACCGGCGACTGTTTAATCTTCTGGTG-3′) using the Phusion DNA polymerase.

Both plasmids were cloned using the infusion cloning system (Takara).

**Co-cultures and in vitro electroporation**
Co-cultures were performed on polylysine/laminin-coated glass cov-erslips, either placed in culture wells or fixed to the bottom of perfo-rated Petri dishes in order to image migrating MGE cells. Brains were collected in cold PBS at embryonic day E14.5. Cortices and MGE explants were then dissected in cold Leibovitz medium (Invitrogen). Cortices were mechanically dissociated. Dissociated cortical cells were cultured on the coated glass coverslips and left in the incubator (37 °C, 5% CO2) during MGE electroporation. For electroporation, each MGE explant was placed in a small well of 3% agar. The desired constructs

(described above) were diluted (1 μg/μl or 0,7 μg/μl) with the pCAGGs-GFP construct (0,5 μg/μl or 0,4 μg/μl) in PBS with Fast Green added at a final concentration of 0,01%. The plasmid solution was micro-injected within the MGE explants using glass capillaries (Narishige, G-1.2). One pulse (100 V, 5 msec) was then delivered with a BTX electroporator using a petri dish equipped with electrodes (Nepagene, Sonidel, UE). Electroporated explants were then left to recover for at least one hour in F12/DMEM medium with 10% calf serum in the incubator (37 °C, 5% CO2). Explants were then divided into smaller pieces and co-cultured on the dissociated cortical cells. MGE explants electroporated with a given construct were co-cultured on the same cortical substrate as MGE explants electro-porated with the corresponding control construct to enable sub-sequent live imaging in the same conditions.

**Brain organotypic slices**
Organotypic slices were prepared from E15.5 embryonic brains embedded in 3% type VII agar (Sigma, A0701) and sectioned cor-onally using a manual slicer into 250 μm thick sections. Slices were then transferred onto Millicell chambers (Merck Millipore) for cul-ture. After electroporation (as described above), E15.5 MGE explants were grafted in the cultured organotypic slices at the pallium/sub-pallium boundary. Grafted slices were then left to culture for 60 h prior to fixation.

**Pharmacology experiments**
Recombinant mouse CXCL12 (Ref 460-SD-010, R&D systems, USA) or forskolin (Ref 11018, Cayman chemical) were diluted in the culture medium and applied on co-cultures at the onset of imaging or after 5 h of imaging by replacing half the volume of culture medium with the drug solution. CXCL12 and forskolin were applied at a final concentration of 0,125 nM and 10 μM, respectively. In the case of cAMP immunostaining, CXCL12 and forskolin were applied 15 min before fixation, at a final concentration of 12,5 nM and 1 μM, respectively.

In the case of the CXCR4 antagonist AMD3100 (Ref A5602-5MG, Sigma), AMD3100 was diluted in the culture medium and applied on co-cultures 30 min before the onset of imaging, at a final concentration of 50 μM. AMD3100 was replaced by a combination of AMD3100 and CXCL12 diluted in the culture medium at a final concentration of 50 μM and 0,125 nM (respectively), and live imaging was initiated.

**Videomicroscopy**
Time-lapse imaging was performed at 37 °C with an inverted micro-scope (Leica DMI4000) equipped with a spinning disk (Roper Scien-tific, USA) and with a temperature-controlled chamber. The co-culture medium was replaced prior to acquisition with a culture medium of the same composition but without phenol red. Multi-position acquisition was performed with a Coolsnap HQ camera (Roper Scientific, USA) to allow the recording in the same conditions of MGE cells electroporated with a given construct or its corresponding control construct. Images were acquired with a x20 objective (LX20, Fluotar, Leica, Germany) and 491 nm and 561 nm lasers (MAG Biosystems, Arizona). Z-stacks of 7 μm were acquired with a step size of 1 μm every minute in the case of bPAC photo-activation or every 5 min for all other electroporated constructs, for up to 15 h. A maximum of three experimental condi-tions (including the control condition) were imaged simultaneously, with a minimum of three positions defined for each experimental condition. Acquisitions were controlled using the Metamorph soft-ware (Roper Scientific, USA).

**Immunohistochemistry experiments**
Co-cultures were fixed after 26 or 48 h of culture in 4% PFA/ 0.33 M sucrose in 0.12 M Phosphate Buffer for 15 min at room temperature. For γ-tubulin staining, co-cultures were fixed for 5 min at room

temperature in 4% PFA/ 0.33 M sucrose in 0.12 M Phosphate Buffer and then for 10 min at −20 °C in 100% methanol. In the case of cAMP immunostaining, cells were fixed in 4% PFA/ 0.33 M sucrose/ Picric acid 1X in 0.12 M Phosphate Buffer for 30 min at room temperature. Organotypic slices were fixed by immersion in cold 4% PFA in 0.12 M Phosphate Buffer for 3 h and then pre-incubated for at least 5 h in PGT (PBS; gelatin 2 g/L; 0,25% Triton X-100).

Organotypic brain slices or co-cultures were then incubated overnight at 4 °C in primary antibodies respectively diluted in PGT or PBT (PBS; 0.25% Triton X-100) with 2% normal goat serum and 1% bovine serum albumin (BSA, Sigma-Aldrich). After 3 rinses with PBT, brain sections or cells were incubated for 2 h or 1h30 (respectively) with secondary antibodies diluted in PBT at 1/400. Brain sections were extensively washed in PBS after antibody incubation.

The following primary antibodies were used: chicken anti GFP (1/500, Aves Lab GFP-1020), rat anti RFP (1/1000, ChromoTek, clone 5F8), rabbit anti Arl13b (1/500, Proteintech 17711-1-AP), mouse anti γ-tubulin (1/4000, Sigma T6557, clone GTU-88), rabit anti cAMP (1/200, Merck 07-1497). Primary antibodies were revealed by immuno-fluorescence with the appropriate Alexa dye (Molecular Probes) or Cy3- and Cy5- conjugated secondary antibodies (Jackson laboratories) diluted in PBT (1/400). Bisbenzimide (1/5000 in PBT for 10 min (co-cultures) or 25 min (organotypic slices), Sigma) was used for nuclear counterstaining.

Co-cultures and brain organotypic slices were mounted in mowiol/DABCO (25 mg/mL) and observed on a macroscope (MVX10 olympus), a LEICA DM6000 upright fluorescent microscope using an immersion x100 objective or on a confocal microscope (Leica TCS SP2) using an immersion x63 objective.

## Image processing

For co-culture experiments, for each migrating cell, cell directionality ratios, as well as the mean migration speed, pause duration and number of translocations per hour were extracted from spinning disk acquisitions using the MTrackJ pluggin of ImageJ (NIH, USA) or the Metamorph software. Only cells co-electroporated with the construct of interest and the cytoplasmic GFP construct were taken into account for the analyses. Cells electroporated with only one of the two constructs – i.e., the PC-targeted sponge or the cytoplasmic GFP construct – were excluded from the analyses, in favour of co-electroporated cells. Directionality ratios are extracted for each cell at each migration time point and correspond to the ratio between the distance of the direct path the cell could have chosen and the distance of the real path it has followed. The duration of persistent migration corresponds to the average migration time spent by a cell without any change in polarity. Pauses were defined as the consecutive imaging time points with instant migration speeds (i.e., occurring between two imaging frames) below 25 μm/h. Mean pause duration corresponds to the mean of all the pausing periods displayed by a cell during its entire migration sequence. For each migrating cell, a translocation event was defined for each instant migration speed with a value higher or equal to 120 μm/h. The frequency of nucleokinesis was defined for each cell as the ratio between the number of translocation events and the time spent by the cell in migration. Average values obtained from three independent experiments for each individual cell were used for statistical analyses.

For ex vivo brain organotypic slice experiments, DAPI staining allowed to distinguish between the VZ/SVZ, IZ and CP regions. The proportion of cells co-electroporated with the PC-targeted scavengers and the cytoplasmic GFP construct was estimated around 90 %. On this basis, analyses were carried out on GFP-positive cells. Leading process orientation was extracted using ImageJ (NIH, USA). Angles obtained from the different slices of four (control 5HT6 condition) or five (5HT6-CAMP Sponge and 5HT6-SponGee conditions) embryonic brains were used for statistical analyses.

## FRET imaging

Images were acquired with an inverted DMI6000B epifluorescence microscope (Leica) coupled to a 40x oil-immersion objective (N.A. 1.3) and Metamorph software (Molecular Devices). Images were acquired simultaneously for the CFP (483/32 nm) and YFP (542/27) channels. Simultaneous CFP and YFP channel acquisition was achieved using a dual chip CCD camera ORCA-D2 (Hamamatsu). The wavelength used for CFP excitation was 436/20 nm. The co-cultures were continuously superfused (0.2 mL.min-1) using a closed chamber (FCS2, Bioptechs) and a syringe-pump (Aladdin, WPI), to avoid imaging artefacts generated by the pulses of peristaltic pumps. The superfusion medium was: 1 mM $CaCl_2$, 0.3 mM $MgCl_2$, 0.5 mM $Na_2HPO_4$, 0.45 mM $NaH_2PO_4$, 0.4 mM $MgSO_4$, 4.25 mM KCl, 14 uM $NaHCO_3$, 120 uM NaCl, 0.0004% $CuSO_4$, 0.124 uM $Fe(NO_3)_3$, 1.5 uM $FeSO_4$, 1.5 uM thymidine, 0.51 uM lipoic acid, 1.5 uM $ZnSO_4$, 0.5 mM sodium pyruvate (all from Sigma), 1X MEM Amino Acids (Life Technologies), 1X non-essential amino acids (Life Technologies), 25 mM HEPES (Sigma), 0.5 mM putrescine (Sigma), 0.01% BSA (Sigma), 0.38% glucose (Sigma), 1 mM glutamine (Life Technologies), 1% penicillin streptomycin (Life Technologies). Vitamin B12 and riboflavin were omitted because of their auto-fluorescence. The acquisition lasted 30 min, registering one image every 20 sec, registering in parallel 6 positions on the same coverslip. 10 min after the beginning of the acquisition, cells were subjected to a 20 s or 40 s pulse of 10uM forskolin (R&D Systems) or 5uM spermine NONOate (Sigma) respectively. 10 min after the first pulse, 10uM forskolin or 50uM spermine NONOate was applied for 10 min.

## FRET data analysis

Images were split in two using ImageJ to separate the CFP from the YFP channel. Data were then analysed by calculating the FRET ratio at each time point for one or several Regions Of Interest (ROIs). The user defined ROIs for each position. For each ROI, the raw FRET ratio was computed as Rc = (ICFP − BCFP) / (IYFP − BYFP) with the cAMP sensor and to Rc = (IYFP − BYFP) / (ICFP − BCFP) with the cGMP sensor, where IYFP is the mean intensity of the ROI in the YFP channel; BYFP is the mean intensity of the background in the YFP channel (measured in an ROI of the image devoid of electroporated cell); ICFP is the mean intensity of the ROI in the CFP channel; BCFP is the value of the background in the CFP channel (measured in an ROI of the image devoid of electroporated cell). The FRET measurements exhibited a drift before the first forskolin or spermine NONOate stimulation. The measurements were corrected for this drift. For each ROI, the slope of the drift was measured by fitting 4 min amongst the 10 first minutes of measurements to a straight line. The slope of this fitted line was then used for drift correction of all FRET ratio time points. The FRET ratio was then set to 0 before forskolin or spermine NONOate stimulation (subtraction of the average of the 5 first minutes of the experiment) and to 100 at the end of the experiment (as average of the 5 last minutes of the experiment). FRET Ratio = 100 x (Rc − R0) / (Rf − R0), where Rc is the value of the drift-corrected FRET ratio, R0 the mean of the baseline and Rf the mean of the plateau. The measure for each cell was evaluated for three criteria: a minimal CFP and YFP initial intensity ( > 30), a maximal standard deviation of the FRET ratio during the 10 first minutes and a minimal elevation of the FRET ratio after the last forskolin or spermine NONOate stimulation (Rf > 3% of R0). Cells that did not meet these criteria were excluded from the analysis.

## Statistical analyses

All data were obtained from at least three independent experiments and are presented as violin plots, in which thick and thin dotted lines represent the median and quartiles (first and third), respectively. Error bars presented in figures are SEM. Statistical analyses were performed with the GraphPad Prism software. Statistical significance of the data was evaluated using the unpaired two-tailed $t$ test, the Mann–Whitney

test, the Chi2 test or the Two-way ANOVA test followed by a Bonferroni post-hoc test. Data distribution was tested for normality using the D'Agostino and Pearson omnibus normality test. Values of $p < 0.05$ were considered significant. In figures, levels of significance were expressed by * (or #) for $P < 0.05$, ** (or ##) for $P < 0.01$, *** (or ###) for $P < 0.001$ and **** (or ####) for $P < 0.0001$.

## Reporting summary

Further information on research design is available in the Nature Portfolio Reporting Summary linked to this article.

## Data availability

The data generated in this study are provided in the Source Data file. Source data are provided with this paper.

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

## Acknowledgements

This work was supported by the "Institut National de la Santé et de la Recherche Médicale" (INSERM, C.M., M.A., M.D.), the "Agence Nationale de la Recherche" (ANR-18-CE16-0017, LocalHubs, C.M., X.N.),the "Ligue Nationale Contre le Cancer" (Subvention de Recherche Scientifique 2022, C.M.), the Brain & Behaviour Research Foundation (Young Investigator Grant, two years, M.A.) and the Jerome Lejeune Foundation and the Sisley-d'Ornano Foundation (two-year postdoctoral fellowship, M.A.). We warmly thank the imaging platform of the "Institut du Fer à Moulin" for the use of their microscopes, as well as Pr. Fujio Murakami and Pr. Fumio Matsuzaki for their kind construct gifts. We are grateful to all members of the Métin team for their support and constructive discussions.

## Author contributions

C.M. designed the study. M.A. performed the experiments and analysed the data. M.W. performed the FRET recording and analysed the FRET data with X.N., M.D, F.R. and X.N. designed the constructs. X.N. had insightful comments on the project and the manuscript. M.A. and C.M. wrote the manuscript.

## Competing interests

The authors declare no competing interests.
