## [Peer Review File · Nature Communications]

CXCL12 targets the primary cilium cAMP/cGMP ratio to regulate cell polarity during migrationREVIEWER COMMENTS

Reviewer #1 (Remarks to the Author):

In this very interesting manuscript, Atkins and colleagues address the question of how cyclic nucleotides in distinct subcellular compartments regulate cell migration. They focus on precursors of cortical inhibitory neurons originating in the medial ganglionic eminence (mGE-cIN). They use genetically encoded chelators of cAMP and cGMP and target these tools either to the entire cytoplasm, the centrosome or the primary cilium (PC). Migration is studied in in vitro settings such as explant and slice cultures.

The authors convincingly demonstrate that buffering cAMP and cGMP at the PC differently affects diverse parameters of cell migration (migration speed, branching behavior, and cell directionality). In contrast, buffering these nucleotides at the centrosome elicits similar, but rather subtle effects on migration speed, pausing behavior and the number of nuclear translocations. Unexpectedly and surprisingly, buffering cAMP and cGMP in the cytoplasm has no effect on migration (speed and directionality).

By selectively increasing cAMP in the PC through targeting light-activated adenylate cyclase toward the cilium, the authors provide findings suggesting that the cAMP/cGMP ratio within the PC is critical for the observed effects of these cyclic nucleotides on migration behavior.

Next, the authors address how buffering of cyclic nucleotides affects the effects of CXCL12 on mGE-cIN. CXCL12 is probably the best-characterized attractant acting on mGE-cIN. For instance, it has been reported that CXCL12 supports layer-specific migration of mGE-cIN in the embryonic cortex by decreasing cAMP. Further, acute CXCL12 application to interneurons increases migration speed and decrease branching frequency (Lysko et al., 2011, 2014). On the other hand, Cxcl12-deficiency increases migration speed (Saaber et al., 2019). Here, the authors confirm that acute CXCL12 treatment increases speed and directionality of control cells. After buffering cAMP in the PC, CXCL12 has no effect on directionality (it does not induce a further increase of directionality which is already very high due to cAMP buffering). However, CXCL12 induces an increase in speed. After cGMP buffering in the PC, CXCL12 increases both parameters.

Finally, by assessing mGE-cIN in cortical slices, authors provide data suggesting that modulating cyclic nucleotides in the PC can induce a switch from tangential to radial migration in cortical tissue.

Comments:

This is a very interesting study using elegant tools to study how cyclic nucleotides in different cellular compartments affect migration of mGE-cIN. To modulate this process, the authors use the CXCL12, CXCR4, ACKR3 signaling pathway, which is considered the most important Gi-regulating pathway in mGE-cIN (Wang et al., 2011) and is important for many other cell types as well.

Criticism:

1. Effectiveness of the cAMP and cGMP chelators in reducing cAMP and cGMP levels was not verified under the experimental condition studied here. Is cAMP reduced in mGE-cIN when using cAMP-Sponge? cAMP biosensors that can be electroporated along with the chelators are available and should be used to validate the tools used here (for instance: use a cAMP sensor along with the cytoplasmic cAMP chelator).

2. At least two CXCL12 receptors are present in mGE-cIN (CXCR4 and ACKR3). Both receptors were reported to modulate migration behavior of mGE-cIN through different downstream signaling pathways (Wang et al., 2011; I admit that there is controversy regarding ACKR3-mediated downstream signaling). Authors assume that their CXCL12 effects are mediated by CXCR4 and I believe that this correct. Still, authors must provide data supporting their assumption: either by using *Cxcr4*-deficient mice or by using a CXCR4 antagonist in combination with CXCL12 and the chelators (care should be taken when choosing the substance: authors should select a pure antagonist and avoid substances for which partial agonistic activity was reported).

3. Authors report that the CXCL12-induced increase in migration speed is most likely not caused by CXCR4 in the PC. So where does CXCR4 act? CXCR4 is present all over the cytoplasm of mGE-cIN (leading process including tip, cell body, and trailing process). I would assume CXCR4 influences migration by decreasing cAMP in the cytoplasm and that the extracellular CXCL12 gradient dictates where the effect within the cell is strongest. This, however, is contradicted by the finding that the cytoplasmic cAMP chelator has no effect on migration parameters. Could it be that the buffering effect of this construct is less efficient than the CXCR4-induced decrease of cAMP in the cytoplasm? See my argumentation in point 1: cAMP levels were not visualized or measured. In the absence of an effect on migration, it remains unclear if the construct efficiently reduces cAMP in mGE-cIN. Authors need to solve which CXCL12 effects are affected when targeting light-activated adenylate cyclase to the PC and, more importantly, when targeting the enzyme to the cytoplasm. The latter experiment is important to validate the conclusion that cAMP levels in the cytoplasm do not affect migration behavior.

4. I do not see data that justify the statement in the abstract that “CXCL12 alters ciliary cAMP/cGMP ratio”.

Minor:

5. Authors should cite findings by live cells imaging in cortical slices:

-Wang et al., (2011): *Cxcr4* and *Ackr3* deficiency reduce speed in the SVZ.

-Saaber et al. (2019): *Cxcl12*-deficiency leads to non-directional migration in the cortical plate (i.e. it increases for- and backward migration in the cortical plate).

6. The authors describe that CXCL12 released from cortical progenitors regulates migration of mGE-cIN in the SVZ and cite a paper by Sessa et al. to support this claim. In this paper, the authors ablated cortical progenitors – so it is not clear if the effect on cIN is through CXCL12 (it could be any mediator generated by these cells). Direct proof for the concept that intermediate progenitors regulate mGE-cIN migration comes from targeted Cxcl12 deletion in intermediate progenitors (Abe et al., 2015).

Thus, I think this is potentially very interesting work not only for researches interested in interneuron development, but for a broad readership interested in signaling and cell migration.

However, there are several severe shortcomings that prevent acceptance at the moment.

Reviewer #2 (Remarks to the Author):

The manuscript by Atkins et al describes a study examining the role of cAMP and cGMP signaling in migrating interneurons. The authors use constructs that scavenge these second messengers to make a case that the ratio of cAMP to cGMP in the primary cilia impacts the polarity and branching in this neuronal cell type and the modulating their levels at the basal body does not. Moreover, they seek to connect signaling from the SDF1 chemokine and chemokine receptors to the ciliary cAMP to cGMP ratios as a means for chemokine control of interneuronal migration. While the tools generated for this study are intriguing, the migration phenotypes are mild, and there are confusing points of this study that render its conclusions too preliminary for publication until resolved by further experiments.

Major points

-A main confusing point in this study is related to the Metin laboratory's early work describing the role of cilia in interneuron migration, where they describe electron microscopy studies that only 30% of interneurons possess primary cilia. 30% of neurons possessing the "steering wheel" open questions for this study. Are cilia-bearing cells the only ones to respond to SDF1 application in their in vitro and ex vivo experimental systems? Are the mild migration phenotypes described in the study due to a minor subpopulation of cells that respond to SDF1 or their cAMP/cGMP sponges (i.e. the cilia-bearing cells) while a larger population of non-cilia-bearing cells that do not? This manuscript is written from the perspective that all cells possess the ciliary steering wheel but needs experimental data at multiple levels to account for the ciliary diversity in this cell population. Indeed, if cilia are such an important steering wheel a revised discussion should also mention how interneurons without cilia find their final destination.

-The authors claim that “CXCL12, therefore, appears sufficient to convert the directionality of a control-like migrating cell to 5HT6-cAMP sponge-like phenotype” without actually showing that application of SDF1 modulates cyclic nucleotide levels in the primary cilia of migrating neurons. Correlation does not always mean causation, and the manuscript in its current form does not show that SDF1 causatively alters ciliary cyclic nucleotide levels in this system which is a requirement of the model. Moreover, the authors claim that CXCR4 GPCR does not directly act in cilia to modulate cyclic nucleotide levels: how this occurs is an open question that would greatly add to the significance of the study.

-The authors claim on page 7, line 265 that their work shows SDF1 is acting as a chemoattractant. Bath application of SDF1 does not create directional gradients needed to assess chemoattractant functions and their relationship to polarity and directionality. The authors should temper their claims to arguments about a motogenic function given their application methods.

Reviewer #3 (Remarks to the Author):

Atkins et al. investigate the migration of cortical interneurons in cell and tissue culture systems. They designed and generated a series of targeted scavengers of cGMP and cAMP and photoactivatable adenylyl cyclase to manipulate cyclic nucleotide concentrations in the primary cilium, at the centrosome, or in the entire cell. The cAMP/cGMP balance inside the primary cilium controls the curvature of the migration path while cell-wide manipulations have no measurable effect on migration. The authors suggest a model where the cytokine CXCL12 binds to the ciliary CXCR4 receptor, reducing ciliary cAMP via Gi-mediated inhibition of adenylyl cyclase, promoting straight (tangential) migration along the SVZ. Cells curve and switch to radial migration once they reach an area of low CXCL12 concentration. The model is novel and provides a compelling interpretation of the experimental data. How the cAMP/cGMP balance is read out, exported and translated to altered cytoskeletal dynamics to affect the direction of migration remains open. (The steering wheel is not connected to the tires.) Nevertheless, this elegant study marks a significant advance in our understanding of localized cyclic nucleotide signaling and the function of the primary cilium in neurons.

Major comments:

- 1) Fig. 3I and J: The metaphorical balances are drawn the wrong way around: In Fig. 3I, cAMP dominates (is heavier), thus the scales should be tipped towards cAMP. It is perhaps safer to get rid of the balance drawings and just use different font sizes to symbolize the different concentrations. Using two different metaphors (steering wheel, weighing balance) for the same thing is anyways not recommended.
- 2) Fig. 4F and G: “The number of cells is indicated below graphs.” This is unusual, provide n in the legend instead. In both panels, I noticed a very strong correlation between the height of the bars and the number of cells analyzed in each group (Panel F, R squared= 0.8; panel G, R squared = 0.9, by my calculation). Is this a freak coincidence (twice!) or is something wrong here?

3) Data presentation (all Figs): Bar plots (mean \pm SEM) make it impossible to judge the distribution of individual measurements. Please show the distributions, e.g. using violin plots. I did read the sentence in the methods about normality tests, but seeing is believing.

Minor:

4) Fig. 1F, last panel: the arrow seems to be misplaced

5) 304: "Data not shown": please show data or remove statement

6) 306: "remain misunderstood" - remain unknown

7) 359: "a ciliary cAMP/cGMP balance inversion" - balance shift?

8) Fig. 6, legend: "Summary and conclusive hypothetical model depicting..." verbose, just say "Model of..."

9) 361: "...the Shh morphogen, previously described..." citation missing

10) (optional) There is a substantial body of work about the cAMP control of the sperm flagellum. This could be referenced in the discussion as another instance of cyclic nucleotides in cellular motion control.

REVIEWER COMMENTS

We wish to thank the reviewers for their comments. Their input was extremely helpful to strengthen our data, whether by the suggestion of additional control experiments, a better representation of our data or by giving us the opportunity to clarify certain approaches and conclusions.

Reviewer #1 (Remarks to the Author):

In this very interesting manuscript, Atkins and colleagues address the question of how cyclic nucleotides in distinct subcellular compartments regulate cell migration. They focus on precursors of cortical inhibitory neurons originating in the medial ganglionic eminence (mGE-cIN). They use genetically encoded chelators of cAMP and cGMP and target these tools either to the entire cytoplasm, the centrosome or the primary cilium (PC). Migration is studied in in vitro settings such as explant and slice cultures.

The authors convincingly demonstrate that buffering cAMP and cGMP at the PC differently affects diverse parameters of cell migration (migration speed, branching behavior, and cell directionality). In contrast, buffering these nucleotides at the centrosome elicits similar, but rather subtle effects on migration speed, pausing behavior and the number of nuclear translocations. Unexpectedly and surprisingly, buffering cAMP and cGMP in the cytoplasm has no effect on migration (speed and directionality).

By selectively increasing cAMP in the PC through targeting light-activated adenylate cyclase toward the cilium, the authors provide findings suggesting that the cAMP/cGMP ratio within the PC is critical for the observed effects of these cyclic nucleotides on migration behavior.

Next, the authors address how buffering of cyclic nucleotides affects the effects of CXCL12 on mGE-cIN. CXCL12 is probably the best-characterized attractant acting on mGE-cIN. For instance, it has been reported that CXCL12 supports layer-specific migration of mGE-cIN in the embryonic cortex by decreasing cAMP. Further, acute CXCL12 application to interneurons increases migration speed and decrease branching frequency (Lysko et al., 2011, 2014). On the other hand, Cxcl12-deficiency increases migration speed (Saaber et al., 2019). Here, the authors confirm that acute CXCL12 treatment increases speed and directionality of control cells. After buffering cAMP in the PC, CXCL12 has no effect on directionality (it does not induce a further increase of directionality which is already very high due to cAMP buffering). However, CXCL12 induces an increase in speed. After cGMP buffering in the PC, CXCL12 increases both parameters.

Finally, by assessing mGE-cIN in cortical slices, authors provide data suggesting that modulating cyclic nucleotides in the PC can induce a switch from tangential to radial migration in cortical tissue.

Comments:

This is a very interesting study using elegant tools to study how cyclic nucleotides in different cellular compartments affect migration of mGE-cIN. To modulate this process, the authors use the CXCL12, CXCR4, ACKR3 signaling pathway, which is considered the most important Gi-regulating pathway in mGE-cIN (Wang et al., 2011) and is important for many other cell types as well.

Criticism:

1. Effectiveness of the cAMP and cGMP chelators in reducing cAMP and cGMP levels was not verified

under the experimental condition studied here. Is cAMP reduced in mGE-cIN when using cAMP-Sponge? cAMP biosensors that can be electroporated along with the chelators are available and should be used to validate the tools used here (for instance: use a cAMP sensor along with the cytoplasmic cAMP chelator). We thank the reviewer for suggesting this control experiment, which is important to strengthen our approach. To answer this point, FRET experiments have been set up and performed in Xavier Nicol's lab. These new results are illustrated in the **Supplementary Fig. 3**. The efficiency of the scavengers was validated by assessing their ability to alter a pharmacologically induced increase of either cAMP or cGMP second messengers, as has previously been performed for cAMP sponge (Lefkimiatis et al., 2009) and SponGee (Ros et al., 2019) in other cell types. The cytoplasmic cAMP or cGMP scavengers were co-electroporated in our MGE explants with the FRET biosensors H147 (for cAMP) or ^THPDE5^{VV} (for cGMP), respectively. The ability of cells to efficiently respond to a forskolin or NO-induced increase of cAMP or cGMP (respectively) was then assessed by FRET imaging. As reported in supplementary Fig.3 and its legend and in lines 185-187; 544-584 of the revised manuscript, our data show that compared to control cells, expression of the cytoplasmic cAMP or cGMP chelators affects the cell response to a forskolin or NO pulse (respectively), confirming the effectiveness of our tools in buffering cAMP or cGMP.

2. At least two CXCL12 receptors are present in mGE-cIN (CXCR4 and ACKR3). Both receptors were reported to modulate migration behavior of mGE-cIN through different downstream signaling pathways (Wang et al., 2011; I admit that there is controversy regarding ACKR3-mediated downstream signaling). Authors assume that their CXCL12 effects are mediated by CXCR4 and I believe that this correct. Still, authors must provide data supporting their assumption: either by using Cxcr4-deficient mice or by using a CXCR4 antagonist in combination with CXCL12 and the chelators (care should be taken when choosing the substance: authors should select a pure antagonist and avoid substances for which partial agonistic activity was reported). We have addressed this important concern by performing bath application of CXCL12 in combination with the CXCR4 antagonist AMD3100. These new results are illustrated in the **Supplementary Fig. 5**. To ease the interpretation of these results, the presentation of the graphs depicting directionality ratios in Fig. 4D and Fig. 4E has been modified (Fig. 4D-E; lines 834-838). As reported in the revised manuscript (lines 277-285; 325; 384; 481-485) and in the legend of the Supplementary Fig.5, AMD3100 application in combination with CXCL12 abolishes the conversion by CXCL12 of the low directionality phenotype associated with a low ciliary cGMP/cAMP ratio (ciliary cGMP buffering) to the high directionality phenotype of cells with a high ciliary cGMP/cAMP ratio (ciliary cAMP buffering), by maintaining a low migration speed and directionality (also observed in the absence of CXCL12 application). We believe that this result greatly adds to the manuscript in that it supports a CXCL12 effect on cell directionality via a CXCR4-mediated modulation of ciliary cAMP levels.

3. Authors report that the CXCL12-induced increase in migration speed is most likely not caused by CXCR4 in the PC. So where does CXCR4 act? CXCR4 is present all over the cytoplasm of mGE-cIN (leading process including tip, cell body, and trailing process). I would assume CXCR4 influences migration by decreasing cAMP in the cytoplasm and that the extracellular CXCL12 gradient dictates where the effect within the cell is strongest. This, however, is contradicted by the finding that the cytoplasmic cAMP chelator has no effect on migration parameters. Could it be that the buffering effect of this construct is less efficient than the CXCR4-induced decrease of cAMP in the cytoplasm? See my argumentation in point 1: cAMP levels were not visualized or measured. In the absence of an effect on migration, it remains unclear if the construct efficiently reduces cAMP in mGE-cIN.

We agree that in light of the absence of an effect of our cytoplasmic chelators on the migration speed and directionality parameters, the question of the efficiency of these tools in our setup is essential. Our FRET experiments now confirm their efficiency in the cytoplasm of migrating cortical interneurons (see point 1) – as is the case for RGC axons (Baudet et al., joined manuscript; Ros et al., 2019; Averaimo et al., 2016) - and strengthen our conclusion that buffering cAMP or cGMP in the whole cytoplasm does not affect migration speed or directionality. We would like to emphasise that the cytoplasm of migrating interneurons contains many different subcompartments, such as the leading process, tip, cell body and trailing process mentioned by the reviewer, and therefore cannot be considered as a compartment of its own, as is the case for the primary cilium. Expressing our scavengers in the whole cytoplasm is a control experiment for “no targeting to any subcellular compartment” (line 177-178). This control is important to validate the compartment specific approach to the PC, that is the focus of our study and that unravels a primary cilium component of the CXCL12/CXCR4 axis on the regulation of cell polarity during migration. The question of the finer understanding of the role of the cytoplasmic CXCL12/CXCR4/cAMP axis on migratory behaviours – and of where the regulation of migration speed might fit in – is a whole new question, that we agree is fascinating. It will however most likely require a subcompartment specific approach (within the cytoplasm), for example by distinguishing between leading process, tip, cell body and trailing process subcompartments – for which specific targeting sequences remain to be established. Additionally, given the well documented cyclic behaviour of migrating cortical interneurons (Tsai et al., 2005; Bellion et al., 2005), the possibility that a temporal pattern of local cAMP production/inhibition might alternate at the cell rear and cell front and be superimposed to the spatial pattern cannot be neglected. If such were the case, buffering cAMP in the whole cytoplasm the whole time, as we have performed in our study, would not be sufficient to mimic the increased migration speed induced by CXCL12, for example.

Authors need to solve which CXCL12 effects are affected when targeting light-activated adenylate cyclase to the PC and, more importantly, when targeting the enzyme to the cytoplasm. The latter experiment is important to validate the conclusion that cAMP levels in the cytoplasm do not affect migration behavior.

The reviewer’s concern highlights a lack of clarity in our conclusions regarding the effect of the cytoplasmic buffering of second messenger signals on migration, which we apologise for and now address by providing additional data as well as by a more explicit formulation of our conclusions. Indeed, contrary to what the results and phrasing of our initial manuscript may suggest, our conclusion is **not** that cAMP levels in the cytoplasm do not affect migratory behaviours. To illustrate this point, we provide new control experiments showing that a pharmacologically- or optogenetically-induced increase in cytoplasmic cAMP levels strongly inhibits migration, by inducing a drastic decrease of both migration speed and distance. These new results are illustrated in the **Supplementary Fig. 4**. Such an effect on migration precludes the use of this approach to study the cytoplasmic role of cAMP on migration parameters such as speed, and is in line with the need for more subtle approaches, most likely at the subcellular level.

To address all these key issues, we have modified the text in the result, discussion and Methods sections (lines 176, 195-201; 361; 422; 476-480) of the revised manuscript in addition to the new data provided (see also the legend of the supplementary Fig.4). In the discussion, we had hypothesised that the CXCL12-mediated effect on migration speed could result from a cytoplasmic component of the CXCL12/CXCR4 pathway. Our hypothesis was based on our observations that buffering ciliary cAMP levels does not increase migration speed, suggesting that increased speed may result from a decrease in cAMP levels locally within a cytoplasmic subcompartment. We feel that this

hypothesis adds a complexity that clouds the take-home message of this paragraph – and study: the role of the ciliary cAMP/cGMP ratio as the steering wheel of migration. For sake of focus and clarity, we propose to withdraw this sentence (lines 377-384).

4. I do not see data that justify the statement in the abstract that “CXCL12 alters ciliary cAMP/cGMP ratio”. We fully agree with the reviewer and have replaced the verb “alters” by the more appropriate “targets” (line 19 of the revised manuscript).

Minor:

5. Authors should cite findings by live cells imaging in cortical slices:

-Wang et al., (2011): Cxcr4 and Ackr3 deficiency reduce speed in the SVZ.

-Saaber et al. (2019): Cxcl12-deficiency leads to non-directional migration in the cortical plate (i.e. it increases for- and backward migration in the cortical plate).

These two publications should have been cited in the initial manuscript, and are even more appropriate in the revised one given the control experiments with the CXCR4 antagonist. We apologise for the oversight and have added them to our manuscript (line 235).

6. The authors describe that CXCL12 released from cortical progenitors regulates migration of mGE-cIN in the SVZ and cite a paper by Sessa et al. to support this claim. In this paper, the authors ablated cortical progenitors – so it is not clear if the effect on cIN is through CXCL12 (it could be any mediator generated by these cells). Direct proof for the concept that intermediate progenitors regulate mGE-cIN migration comes from targeted Cxcl12 deletion in intermediate progenitors (Abe et al., 2015).

This is an important point as our model relies on this process. We apologise for the inaccuracy and omission and have cited the publication in the manuscript line 297.

Thus, I think this is potentially very interesting work not only for researches interested in interneuron development, but for a broad readership interested in signaling and cell migration.

However, there are several severe shortcoming that prevent acceptance at the moment.

Reviewer #2 (Remarks to the Author):

The manuscript by Atkins et al describes a study examing the role of cAMP and cGMP signaling in migrating interneurons. The authors use constructs that scavenge these second messengers to make a case that the ratio of cAMP to cGMP in the primary cilia impacts the polarity and branching in this neuronal cell type and the modulating their levels at the basal body does not. Moreover, they seek to connect signaling from the SDF1 chemokine and chemokine receptors to the ciliary cAMP to cGMP ratios as a means for chemokine control of interneuronal migration. While the tools generated for this study are intriguing, the migration phenotypes are mild, and there are confusing points of this study that render its conclusions too preliminary for publication until resolved by further experiments.

Major points

-A main confusing point in this study is related to the Metin laboratory's early work describing the role of cilia in interneuron migration, where they describe electron microscopy studies that only 30% of interneurons possess primary cilia. 30% of neurons possessing the "steering wheel" open questions for this study. Are cilia-bearing cells the only ones to respond to SDF1 application in their in vitro and ex vivo experimental systems? Are the mild migration phenotypes described in the study due to a minor subpopulation of cells that respond to SDF1 or their cAMP/cGMP sponges (i.e. the cilia-bearing cells) while a larger population of non-cilia-bearing cells that do not? This manuscript is written from the perspective that all cells possess the ciliary steering wheel but needs experimental data at multiple levels to account for the ciliary diversity in this cell population. Indeed, if cilia are such an important steering wheel a revised discussion should also mention how interneurons without cilia find their final destination.

The concern raised by the reviewer is a very important one, on which we evidently do not sufficiently elaborate in the initial manuscript. We thank the reviewer for giving us the opportunity to rectify the situation. As now reported in the revised manuscript (lines 46-47; 112-114) and as is visible in the Supplementary Movies 1 and 2, the primary cilium is a highly dynamic organelle, that the cell rhythmically extends and retracts from its surface. As a consequence, at a given time point located between two cycles of ciliogenesis, a migrating cell may appear non ciliated, until re-entering a cycle of ciliogenesis and extending a new PC at its surface at a later time point. For this reason, fixed preparations as the one mentioned by the reviewer in a previous study from the lab (Baudoin et al., 2012) are ill-adapted to extrapolate a proportion of ciliated cells. Nevertheless, with regards to this study, the statement that it reports that "only 30% of interneurons possess primary cilia" is inaccurate. As described in Fig. 1L, the study distinguishes two categories of cells, those with a centrosome located close or at a distance from the nucleus. For each cell, the presence of a primary cilium or of a ciliary vesicle – sign of an early phase of ciliogenesis (Zhao et al., 2023) – was quantified. The "third of [these] cells" with a primary cilium refers to the category of cells exhibiting a primary cilium located far away from the nucleus and does not include the cells with a primary cilium close to the nucleus or those with a ciliary vesicle, which are also destined to form a primary cilium. In this study, on a total of 84 analysed cells, 43 displayed either a primary cilium or a ciliary vesicle, bringing to 50% the proportion of ciliated cells in this fixed preparation. It is interesting to point out that in a later study from the lab using these electron microscopy preparations, 67% of the cells were ciliated (Luccardini et al., 2015; Fig. 4D). These observations provide solid ground to assume that a great majority if not all of our migrating interneurons are ciliated.

Most importantly, as now further developed in the result section (lines 114-115), in addition to the result (lines 111-112) and Method section in the initial manuscript (lines 524-527), all our analyses have been carried out dynamically using videomicroscopy and exclusively on ciliated cells – i.e., cells selected on the basis of a visible primary cilium because electroporated with the mRFP-tagged ciliary construct of interest. This rules out the concern that our phenotypes may be diluted by a larger population of non-cilia-bearing cells. We would like to emphasise that, however subtle, buffering ciliary cAMP is sufficient to mimic the increased directionality and decreased branching induced by CXCL12, which is the most studied chemokine involved in cortical interneuron migration. Moreover, defective primary cilium function has been reported as the cause of a group of developmental disorders termed ciliopathies, which have been associated with defective neuronal migration (Higginbotham et al., 2012; Guo et al., 2015). The fact that there are ciliopathy-affected patients implies that the developmental phenotypes associated with ciliary dysfunction and leading to these

pathologies are sufficiently subtle for patients to be viable and develop the pathology. Better characterising the apparently subtle defects associated with ciliary dysfunction is crucial for our better understanding of such pathologies.

-The authors claim that “CXCL12, therefore, appears sufficient to convert the directionality of a control-like migrating cell to 5HT6-cAMP sponge-like phenotype” without actually showing that application of SDF1 modulates cyclic nucleotide levels in the primary cilia of migrating neurons. Correlation does not always mean causation, and the manuscript in its current form does not show that SDF1 causatively alters ciliary cyclic nucleotide levels in this system which is a requirement of the model.

The reviewer makes a very good point, and we agree that directly measuring changes in cAMP and cGMP levels within the primary cilium of migrating interneurons exposed to CXCL12 would add greatly to the study. Such FRET experiments would imply measuring FRET ratios within a very small volume – cortical interneuron primary cilia are only a few micrometres long and wide –, which in itself would be a challenge. More troubling is the fact that in our system, the primary cilium, as explained in point 1, is highly dynamic, in four dimensions (t,x,y,z; see Supplementary Movies 1 and 2):

- In time, the primary cilium dynamically extends and retracts at the cell surface, implying that it continuously and rapidly varies in length and that it can be absent from the cell surface for stretches of time before re-appearing.
- The primary cilium does not always bud from the same point at the cell surface and moves laterally in the plasma membrane. As it is assembled from the mother centriole, it follows the centrosome, and can bud from the soma – rear or front – surface, the swelling surface... from one cycle of ciliogenesis to another.

For all these reasons, this experiment, although very interesting, is not feasible with the current technology and molecular tools available. Although FRET imaging has already been performed in primary cilia using epithelial cell lines (Jiang et al., 2019; Moore et al., 2016; Delling et al., 2013; Sherpa et al., 2019), such measurements in the primary cilium using neuronal primary cultures, in which the centrosome and primary cilium are highly dynamic, has – to our knowledge – never been achieved. We have nevertheless sought to strengthen our manuscript by carrying out FRET experiments and measuring cAMP and cGMP levels in the whole cytoplasm of our migrating cells, so as to validate the efficiency of our molecular tools. These new results are illustrated in the **Supplementary Fig. 3**.

Moreover, the authors claim that CXCR4 GPCR does not directly act in cilia to modulate cyclic nucleotide levels: how this occurs is an open question that would greatly add to the significance of the study.

We are a little surprised by this comment as in the model we propose (Fig. 6), a ciliary CXCR4-mediated effect of CXCL12 on the ciliary cGMP/cAMP ratio and subsequent polarity of migrating cells is clearly illustrated. We moreover write in the results section that “the CXCR4 receptor for CXCL12 has been found in cortical interneuron primary cilia” (line 246-247) and that “our results support a new conceptual model in which CXCL12 secreted by SVZ cortical progenitors binds to ciliary-located CXCR4 receptors on tangentially migrating MGE cells” (lines 313-314).

-The authors claim on page 7, line 265 that their work shows SDF1 is acting as a chemoattractant. Bath application of SDF1 does not create directional gradients needed to assess chemoattractant

functions and their relationship to polarity and directionality. The authors should temper their claims to arguments about a motogenic function given their application methods.

We fully agree with the reviewer that chemoattractant is inappropriate given our bath application method and have replaced the term by “chemokine” (line 288 of the revised manuscript).

Reviewer #3 (Remarks to the Author):

Atkins et al. investigate the migration of cortical interneurons in cell and tissue culture systems. They designed and generated a series of targeted scavengers of cGMP and cAMP and photoactivatable adenylyl cyclase to manipulate cyclic nucleotide concentrations in the primary cilium, at the centrosome, or in the entire cell. The cAMP/cGMP balance inside the primary cilium controls the curvature of the migration path while cell-wide manipulations have no measurable effect on migration. The authors suggest a model where the cytokine CXCL12 binds to the ciliary CRCR4 receptor, reducing ciliary cAMP via Gi-mediated inhibition of adenylyl cyclase, promoting straight (tangential) migration along the SVZ. Cells curve and switch to radial migration once they reach an area of low CXCL12 concentration. The model is novel and provides a compelling interpretation of the experimental data. How the cAMP/cGMP balance is read out, exported and translated to altered cytoskeletal dynamics to affect the direction of migration remains open. (The steering wheel is not connected to the tires.) Nevertheless, this elegant study marks a significant advance in our understanding of localized cyclic nucleotide signaling and the function of the primary cilium in neurons.

Major comments:

1) Fig. 3I and J: The metaphorical balances are drawn the wrong way around: In Fig. 3I, cAMP dominates (is heavier), thus the scales should be tipped towards cAMP. It is perhaps safer to get rid of the balance drawings and just use different font sizes to symbolize the different concentrations. Using two different metaphors (steering wheel, weighing balance) for the same thing is anyways not recommended.

We thank the reviewer for pointing out this mistake, which had escaped our notice and is at odds with our message and conclusion. In Fig. 3I, J and Fig. 6, we have tipped the scales in combination with different font sizes so that the second messenger with the highest levels appears both heavier and dominant.

2) Fig. 4F and G: “The number of cells is indicated below graphs.” This is unusual, provide n in the legend instead.

We agree with the reviewer. In all figures, the number of cells is now indicated in the graph legend.

In both panels, I noticed a very strong correlation between the height of the bars and the number of cells analyzed in each group (Panel F, R squared= 0.8; panel G, R squared = 0.9, by my calculation). Is this a freak coincidence (twice!) or is something wrong here?

We thank the reviewer for pointing this out to us. We have checked the data but the height of the bars in both panels do correspond to the measured directionality ratios (Fig. 4F) and speed (Fig. 4G), and not to the number of cells indicated below.

3) Data presentation (all Figs): Bar plots (mean +- SEM) make it impossible to judge the distribution of individual measurements. Please show the distributions, e.g. using violin plots. I did read the sentence in the methods about normality tests, but seeing is believing.

We agree with the reviewer that seeing the distributions is important. As suggested, all bar plots have been replaced by violin plots.

Minor:

4) Fig. 1F, last panel: the arrow seems to be misplaced

The arrow was indeed misplaced. We thank the reviewer for pointing it out and have rectified the situation.

5) 304: "Data not shown": please show data or remove statement

The statement has been removed from the revised manuscript (line 336).

6) 306: "remain misunderstood" - remain unknown

We have replaced "misunderstood" by the more appropriate "unknown" (line 338).

7) 359: "a ciliary cAMP/cGMP balance inversion" - balance shift?

We agree with the reviewer and have used "balance shift", as suggested (line 393).

8) Fig. 6, legend: "Summary and conclusive hypothetical model depicting..." verbose, just say "Model of..."

The reviewer's suggestion is indeed much more straightforward and has been applied line 868.

9) 361: "...the Shh morphogen, previously described..." citation missing

We thank the reviewer for pointing out this omission. The corresponding citation is now referred to in the text line 396.

10) (optional) There is a substantial body of work about the cAMP control of the sperm flagellum. This could be referenced in the discussion as another instance of cyclic nucleotides in cellular motion control.

The studies referred to by the reviewer are a very good example of the control of cell motility by second messenger signalling. However, the sperm flagellum falls into the category of motile cilia, which help cells or fluid to move. By opposition, the primary cilium present at the surface of migrating cortical interneurons belongs to the category of non motile cilia, the sensory function of which has historically remained underestimated for many years after their discovery. The focus of our study being the primary cilium, we feel that including this bibliography might bring a certain confusion to our message, which we wish to avoid.

REVIEWER COMMENTS

Reviewer #1 (Remarks to the Author):

Dear authors

My criticism was fully addressed.

I congratulate to this nice piece of work!

In some places you could increase the flow of your narrative: you might want to go through your Results section and remove some statements that refer to findings that you described in a previous paragraph. Provide a straight description of the new findings and - after that - compare them to what you have seen before and how it fits together. It might be easier for the reader to follow.

Reviewer #2 (Remarks to the Author):

The revised manuscript and reviewer response of Atkins et al. provide orthogonal data and argumentation that sides step the core of what would convince me of the broad claims of their paper that are clearly central to the impact of the study. Given how controversial claims of signaling events specifically occurring at the primary cilia at the exclusion of all other cellular sites have been in the past, greater care is needed to overcome this study's limitations.

Major points:

1) The authors specifically claim that ciliary signaling, specifically via Cxcr4 receptors in the primary cilia, directs the polarity of migration in their neuronal system. Having NOT shown that Sdf1 application induces cyclic nucleotide signaling specifically in primary cilia creates a situation where the actual site of the author's claimed signaling event is unclear: it could be anywhere where Cxcr4 is recruited to the plasma membrane (a fact that a cytoplasmic biosensor does not dispel). Without such data, the work is still too preliminary for publication.

2) I appreciate that the authors have clarified the degree of ciliation of their cell type and provided some insight regarding subpopulations in their experimental setup. Given that the group had an inclusion criterion for imaging cells via time-lapse microscopy, this brings up an excellent opportunity for a proper negative control that needs to be included in their work: namely, assaying the polarity of migration of non-ciliated cells for a subset of their experiments. If cilia are the central steering wheel, as the authors propose, then minority non-ciliated cells should have behavior in response to Cxcr4 that differs from the ciliated cells. Such a negative control is superior to, for example, ablating cilia via Kinesin or IFT manipulation since the lack of cilia is a naturally occurring consequence of the cilia cycle and not a perturbation of vital motor components that can have unexpected off-target effects.

Reviewer #3 (Remarks to the Author):

In the revised MS, the authors have addressed my concerns which were mostly about data presentation. Please state somewhere what the lines in the violin plots depict (mean +/- SEM or median +/- quartiles). Other than that, I have no further comments.

REVIEWER COMMENTS

Reviewer #1 (Remarks to the Author):

Dear authors

My criticism was fully addressed.
I congratulate to this nice piece of work!

We thank the Reviewer for his kind assessment of our work.

In some places you could increase the flow of your narrative: you might want to go through your Results section and remove some statements that refer to findings that you described in a previous paragraph. Provide a straight description of the new findings and - after that - compare them to what you have seen before and how it fits together. It might be easier for the reader to follow.

We thank the reviewer for his constructive comment. Statements have been removed from the Result section (lines 233-237; 303-305) to meet the reviewer's concern.

Reviewer #2 (Remarks to the Author):

The revised manuscript and reviewer response of Atkins et al. provide orthogonal data and argumentation that sides step the core of what would convince me of the broad claims of their paper that are clearly central to the impact of the study. Given how controversial claims of signaling events specifically occurring at the primary cilia at the exclusion of all other cellular sites have been in the past, greater care is needed to overcome this study's limitations.

Major points:

1) The authors specifically claim that ciliary signaling, specifically via Cxcr4 receptors in the primary cilia, directs the polarity of migration in their neuronal system. Having NOT shown that Sdf1 application induces cyclic nucleotide signaling specifically in primary cilia creates a situation where the actual site of the author's claimed signaling event is unclear: it could be anywhere where Cxcr4 is recruited to the plasma membrane (a fact that a cytoplasmic biosensor does not dispel). Without such data, the work is still too preliminary for publication.

We agree that the location of the CXCL12/CXCR4 signalling event responsible for the increased cell directionality is a key feature of our study.

As pointed out by the reviewer, CXCL12/CXCR4-mediated signalling events can occur wherever CXCR4 is recruited at the plasma membrane, ciliary or non ciliary. Indeed, CXCR4 has been shown to be recruited at the ciliary surface – in addition to the rest of the cell's surface –, including in cortical interneuron primary cilia (Higginbotham et al., 2012; Monaco et al., 2019). To assess the capacity of CXCL12 to decrease cAMP levels in the primary cilium, we performed cAMP immunostaining experiments (Trousse et al., 2001; Ogata et al., 2012; Luo et al., 2014; Zhou et al., 2016; Cheng et al., 2020) in the presence or absence of CXCL12. Results showed a reduction in the number and fluorescence intensity of cAMP hotspots detected in cortical interneuron primary cilia (Fig. S5), as previously shown by elisa and colorimetric approaches for the cytoplasm (Cabarello et al., 2019; Dwinell et al., 2004).

We next sought to determine whether CXCL12-mediated cyclic nucleotide signalling within the primary cilium is responsible for the increased directionality phenotype we observe. Since CXCR4 receptors cannot be pharmacologically inhibited selectively at the ciliary surface, we alternatively took advantage of the optogenetic adenylyl cyclase bPAC tool to block the CXCL12/CXCR4-mediated inhibition of cAMP production specifically at the primary cilium. Indeed, while CXCL12 induces a Gi-mediated inhibition of transmembrane adenylyl cyclases, this inhibition does not extend to bPAC, which is a bacterial soluble adenylyl cyclase. bPAC activity is tightly correlated with the state of its light-sensitive domain and is thus not affected by mammalian G proteins (Stier et al., 2011). Our results show that when CXCL12 is allowed to inhibit cAMP production at the whole cell surface except in the primary cilium, where the ciliary cGMP/cAMP ratio is blocked in a low configuration that cannot be modulated by CXCL12, the chemokine no longer elicits its response on cell directionality (Fig. 5K-O & Fig. S7). Such data rule out the possibility of CXCL12 acting on the directionality of migrating cells through an extra ciliary CXCL12/CXCR4 signalling mechanism that is independent of the primary cilium cGMP/cAMP ratio. Most importantly, these results identify the modulation of ciliary cAMP levels (and more precisely, a decrease in ciliary cAMP levels) as an absolute requirement for the chemokine's effect on cell polarity during migration.

2) I appreciate that the authors have clarified the degree of ciliation of their cell type and provided some insight regarding subpopulations in their experimental setup. Given that the group had an inclusion criterion for imaging cells via time-lapse microscopy, this brings up an excellent opportunity for a proper negative control that needs to be included in their work: namely, assaying the polarity of migration of non-ciliated cells for a subset of their experiments. If cilia are the central steering wheel, as the authors propose, then minority non-ciliated cells should have behavior in response to Cxcr4 that differs from the ciliated cells. Such a negative control is superior to, for example, ablating cilia via Kinesin or IFT manipulation since the lack of cilia is a naturally occurring consequence of the cilia cycle and not a perturbation of vital motor components that can have unexpected off-target effects.

As correctly pointed out by the reviewer, we did use an inclusion criterion on which to base our analyses, namely cells in which both the cytoplasmic GFP and our different RFP-tagged ciliary constructs had been co-electroporated. This inclusion criterion enabled access to cell shape while guaranteeing the expression of the desired ciliary construct. As a result of this co-electroporation, cells may express throughout the recording session only the GFP construct, only the RFP-tagged ciliary construct, both constructs, or none. The absence of expression of the ciliary construct in a given cell most likely reflects an absence of electroporation, which is independent of the ability of the cell to extend a primary cilium. Just as some cells will express only the RFP-tagged ciliary construct without the cytoplasmic construct – although they do have a cytoplasm –, some cells will express the GFP construct without the ciliary RFP-tagged construct – although they do dynamically extend a primary cilium. Therefore, we have no means to identify non-ciliated cells in our system, and unfortunately, the “proper negative control” does therefore not exist in our system.

Despite the lack of an internal negative control, the question raised by the reviewer remains very interesting to assess the role of the primary cilium compartment as the steering wheel of

migration. Do non ciliated cells still respond to CXCL12? While not as elegant as the genetic tools used so far in the study, the acute genetic ablation of the primary cilium is, in our system, the only way to address the reviewer's question by inducing a population of non-ciliated cells. As now illustrated in the Supplementary Figure 6, *Kif3a*^{-/-} cells show no change in directionality in response to CXCL12 bath application. The absence of response to CXCL12 in the absence of a primary cilium highlights the requirement of the ciliary compartment for CXCL12 to regulate cell polarity during migration.

Reviewer #3 (Remarks to the Author):

In the revised MS, the authors have addressed my concerns which were mostly about data presentation. Please state somewhere what the lines in the violin plots depict (mean +/- SEM or median +/- quartiles). Other than that, I have no further comments.

We thank the reviewer for his comment. A sentence has been added to the Methods section to indicate what the lines in the violin plots depict (lines 600-601).

REVIEWERS' COMMENTS:

Reviewer #2 (Remarks to the Author):

The authors provide new data supporting their model. While they mentioned there are no perfect controls or tests related to my queries, they made a good-faith effort to address my two remaining concerns substantively. I'm confident the manuscript is now in a sound state where the neuronal migration field now has the chance to assess the utility of the model proposed by the authors.

REVIEWERS' COMMENTS:

Reviewer #2 (Remarks to the Author):

The authors provide new data supporting their model. While they mentioned there are no perfect controls or tests related to my queries, they made a good-faith effort to address my two remaining concerns substantively. I'm confident the manuscript is now in a sound state where the neuronal migration field now has the chance to assess the utility of the model proposed by the authors.

We thank the reviewer for his assessment of our work.